# Pseudoexfoliation syndrome-associated genetic variants affect transcription factor binding and alternative splicing of *LOXL1*

Francesca Pasutto[1,*], Matthias Zenkel[2,*], Ursula Hoja[2], Daniel Berner[2], Steffen Uebe[1], Fulvia Ferrazzi[1], Johannes Schödel[3], Panah Liravi[2], Mineo Ozaki[4], Daniela Paoli[5], Paolo Frezzotti[6], Takanori Mizoguchi[7], Satoko Nakano[8], Toshiaki Kubota[8], Shinichi Manabe[9], Erika Salvi[10], Paolo Manunta[11], Daniele Cusi[12], Christian Gieger[13], Heinz-Erich Wichmann[13], Tin Aung[14], Chiea Chuen Khor[15], Friedrich E. Kruse[2], André Reis[1] & Ursula Schlötzer-Schrehardt[2]

Although lysyl oxidase-like 1 (*LOXL1*) is known as the principal genetic risk factor for pseudoexfoliation (PEX) syndrome, a major cause of glaucoma and cardiovascular complications, no functional variants have been identified to date. Here, we conduct a genome-wide association scan on 771 German PEX patients and 1,350 controls, followed by independent testing of associated variants in Italian and Japanese data sets. We focus on a 3.5-kb four-component polymorphic locus positioned spanning introns 1 and 2 of *LOXL1* with enhancer-like chromatin features. We find that the rs11638944:C > G transversion exerts a *cis*-acting effect on the expression levels of *LOXL1*, mediated by differential binding of the transcription factor RXRα (retinoid X receptor alpha) and by modulating alternative splicing of *LOXL1*, eventually leading to reduced levels of *LOXL1* mRNA in cells and tissues of risk allele carriers. These findings uncover a functional mechanism by which common noncoding variants influence *LOXL1* expression.

[1] Institute of Human Genetics, Universitätsklinikum Erlangen, Friedrich-Alexander-Universität Erlangen-Nürnberg, Schwabachanlage 10, 91054 Erlangen, Germany. [2] Department of Ophthalmology, Universitätsklinikum Erlangen, Friedrich-Alexander-Universität Erlangen-Nürnberg, Schwabachanlage 6, 91054 Erlangen, Germany. [3] Department of Nephrology and Hypertension, Universitätsklinikum Erlangen, Friedrich-Alexander-Universität Erlangen-Nürnberg, Ulmenweg 18, 91054 Erlangen, Germany. [4] Ozaki Eye Hospital, 1-15 Kamezaki, Hyuga, Miyazaki 883-0066, Japan. [5] Ospedale Monfalcone, Centro Glaucomi, Via Galvani 1, 34074 Monfalcone, Italy. [6] Ophthalmology Unit, Department of Medicine, Surgery and Neuroscience, University of Siena, Viale Bracci SNC, 53100 Siena, Italy. [7] Mizoguchi Eye Clinic, 6-13 Tawara-machi, Sasebo, Nagasaki 857-0016, Japan. [8] Department of Ophthalmology, Oita University, Faculty of Medicine, 1-1 Idaigaoka, Hasana-machi, Oita 879-5593, Japan. [9] Hayashi Eye Hospital, 4-23-35 Hakataekimae, Hakata-ku, Fukuoka 812-0011, Japan. [10] Department of Health Sciences, University of Milano, Via Ortles 22/4, 20139 Milano, Italy. [11] Department of Nephrology, University Vita-Salute San Raffaele, Via Olgettina 58, 20132 Milano, Italy. [12] Institute of Biomedical Technologies, National Research Centre (ITB-CNR), Via Fratelli Cervi 93, 20090 Segrate-Milano, Italy. [13] Institute of Epidemiology, Helmholtz Center Munich, Ingolstädter Landstr. 1, 85764 Munich, Germany. [14] Singapore Eye Research Institute, Singapore National Eye Center, 11 Third Hospital Avenue, Singapore 168751, Singapore. [15] Genome Institute of Singapore, 60 Biopolis Street, Genome Building #02-01, Singapore 138672, Singapore. * These authors contributed equally to this work. Correspondence and requests for materials should be addressed to U.S.-S. (email: Ursula.schloetzer-schrehardt@uk-erlangen.de).

Glaucoma is one of the leading causes of irreversible blindness worldwide[1]. It represents a neurodegenerative disease complex comprising heterogeneous subtypes that share a common pathogenic pathway, that is, progressive loss of retinal ganglion cells and optic nerve axons resulting in visual field defects. Within this multifarious disease spectrum, pseudo-exfoliation (PEX) glaucoma is considered a frequent and progressive subtype accounting for 25–70% of open-angle glaucoma and involving a high risk of blindness[2,3]. It is thought to be caused by the obstruction of the aqueous humour outflow pathways because of the deposition of extracellular protein aggregates leading to intraocular pressure elevation and subsequent glaucomatous optic nerve damage. This abnormally produced material appears in the course of a common age-related, generalized disorder of the extracellular matrix termed PEX syndrome, which affects up to 30% of the elderly population[4,5]. Based on the systemic nature of the underlying connective tissue disorder, PEX syndrome has been associated not only with glaucoma but also with cardiovascular diseases including cardiomyopathy and abdominal aortic aneurysms[6–8].

Although the incidence of PEX syndrome is clearly influenced by environmental factors[9], there is a strong genetic component to the disease[10]. A genome-wide association study in Scandinavian populations identified lysyl oxidase-like 1 (LOXL1) on chromosome 15q24.1 as a principal genetic risk factor for PEX syndrome/glaucoma[11], a finding that has been replicated in multiple populations worldwide[12–16]. Two common non-synonymous protein-coding variants in exon 1, rs1048661G > T (Arg141Leu) and rs3825942G > A (Gly153Asp), and one intronic variant (rs2165241T > C), conferring a 20-fold increased risk for PEX, were initially considered as the causal variants in this susceptibility locus. However, risk alleles were reversed between Caucasians and other ethnic populations, that is, Asian and black South African populations, and—although a biological effect on LOXL1 protein processing was suggested[17]—the exact functional roles of these missense variants in the pathogenesis of PEX have not yet been ascertained[18]. 'Flipping' of risk alleles and lack of compelling evidence for their functionality prompted further search and led to identification of additional PEX-associated risk variants, rs16958477 and rs12914489, in the LOXL1 promoter region[19–21]. Recently, Hauser and colleagues identified PEX-associated variants at the exon 1–intron 1 boundary of LOXL1 in black South African, US Caucasian, German and Japanese patients, which were suggested to modulate the expression of LOXL1 antisense RNA 1 (LOXL1-AS1), a long non-coding RNA of LOXL1 (ref. 22). However, these variants had no effect on LOXL1 expression which is known to be markedly dysregulated in tissues of PEX patients[23–26]. Collectively, a substantial gap remains between the numerous reports on PEX-associated sequence variants and our understanding of how these variants contribute to disease.

LOXL1 encodes a member of the lysyl oxidase family of enzymes (LOX, LOXL 1 to 4), which catalyses the generation of lysine-derived cross-links in extracellular matrix molecules such as collagen and elastin[27]. The currently best known function of LOXL1 is cross-linking of tropoelastin monomers into elastin polymers during the formation and maintenance of elastic fibres[28]. Current evidence supports a fundamental role for both LOX and LOXL1 in connective tissue homeostasis and stability. Their dysregulated expression has been linked to both fibrotic and elastotic-degenerative connective tissue disorders including lung emphysema, aneurysms, and pelvic organ prolapse[29–32]. Dysregulated expression of LOXL1 also appears to play a key role in PEX pathogenesis[23–26]. Notably, reduced expression of LOXL1 in elastin-rich, load-bearing tissues such as the lamina cribrosa has been suggested as a major susceptibility factor for PEX

glaucoma because of the accompanying elastotic and biomechanical tissue alterations[24,33], and may be also a predisposing factor for cardiovascular complications including aortic aneurysms in PEX patients[6–8]. Thus, deciphering the mechanisms of LOXL1 regulation is vital to understanding the aetiology of PEX syndrome and its potentially sight- and life-threatening complications.

Emerging evidence suggests that intronic variants may have a role in common disease susceptibility by influencing transcriptional output of gene expression[34]. The aim of this study was to identify potential regulatory variants in the LOXL1 locus and functionally characterize their impact on LOXL1 expression regulation. On the basis of a genome-wide association study in a German cohort of PEX patients we select a cluster of 14 common SNPs within introns 1 and 2 of LOXL1 in complete linkage disequilibrium (LD) with known variants, and confirm their association with PEX in European and Asian populations. Using models of disease-relevant cell types, we provide experimental evidence for a functional PEX-associated variant, rs11638944:C > G. Located in a genomic region with regulatory potential downstream of the canonical LOXL1 promoter, this variant exerts allele-specific effects on LOXL1 expression through differential transcription factor binding and alternative pre-mRNA splicing in a cell type-specific manner. Here, we show that increased transcriptional activity at the risk sequence is associated with reduced binding of retinoid X receptor alpha (RXRα) and with enhanced alternative splicing coupled with nonsense-mediated decay (NMD), which altogether reduces the levels of LOXL1 mRNA in cells and tissues of risk allele carriers, underlining a functional link between LOXL1 genetic variation and regulation of LOXL1 expression.

## Results

**Selection of LOXL1 candidate variants.** To capture potentially functional regulatory variants in the LOXL1 gene locus, we first performed a genome wide association study using DNA samples from a German cohort of patients with PEX syndrome/glaucoma (n = 771) and healthy subjects (n = 1,350). As expected, we detected multiple SNPs in high LD at the LOXL1 locus on chromosome 15 showing strong evidence for association with PEX, with rs2028386 as example for main association signal (P = 7.62E − 22, Fig. 1). Within this locus spanning ∼19 kb around exon 1, intron 1, exon 2 and intron 2 of the LOXL1 gene (chr15:74,218,178–74,237,946, GRCh37/hg19), a total of 46 SNPs showed genome wide significant association (P < 10^−8) with PEX (Fig. 2a). These 46 SNPs include the two non-synonymous variants rs1048661 (p.(Arg141Leu)) and rs3825942 (p.(Gly153Glu)) and the intronic variant rs2165241 identified in the original publication by Thorleifsson et al.[11]

Out of the 46 SNPs, we focused on a cluster of 14 SNPs showing the most significant P-values (P < 10^−15) (Fig. 2a, Supplementary Table 1). These 14 SNPs include the known intronic index SNP rs2165241 and are distributed throughout intron 1 and the start of intron 2 of LOXL1 (Fig. 2b). In addition, according to the UCSC genome database, some of these SNPs are positioned within putative regulatory regions (Fig. 2b), suggesting that the functional risk variants may reside within this block. Haplotype analysis including the two index non-synonymous SNPs confirmed that they are located on the same risk haplotype (previously defined by the two non-synonymous and the intronic index SNPs), which is also the more frequent one (ca. 73%) in the German population (Supplementary Table 2).

Analysis of the same LOXL1 region in an Italian cohort (421 cases and 1,505 controls) confirmed significant association for these 14 SNPs with PEX syndrome/glaucoma (Table 1,

Supplementary Table 1, Supplementary Fig. 1). Allelic prevalence of the PEX-associated risk alleles was about 80% in cases and about 40% in controls. In addition, all 14 SNPs were significantly

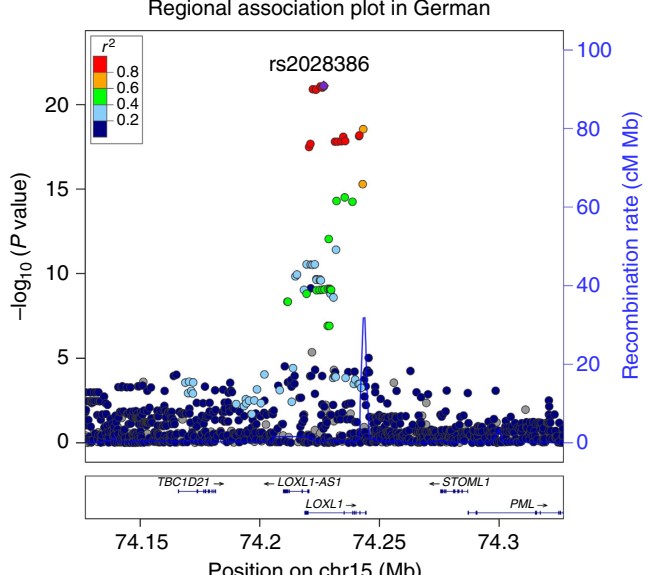

**Figure 1 | Regional association plot of the 15q24.1 locus in GWA analysis of German PEX data set.** Regional association plot for *LOXL1* gene with 100 kb upstream and downstream regions shows the association peak region for SNPs located on the *LOXL1* gene locus. Association analysis was performed using a logistic regression model adjusted for age and gender. The data of all SNPs are indicated as circles. The red and orange circles represent the 14 SNPs showing major association. The left Y-axis represents $-\log_{10} P$ values and the right Y axis represents the recombination rate. The X axis represents position of SNPs on chromosome 15 (human genome build GRCh37/hg19).

associated with PEX syndrome/glaucoma in a Japanese cohort (1,484 cases and 1,188 controls), although risk alleles were completely reversed between the two European and Japanese populations (Table 1). However, in all three cohorts analysed, the 14 selected variants are located within the most frequent haplotype representing the risk haplotype in each specific population as exemplified for the German and Italian cohorts (haplotype 4, Supplementary Table 2).

Conditional analysis to test for independent association between the selected SNPs and the two non-synonymous index SNPs, rs3825942 and rs1048661, revealed no independent association signal among the 14 SNPs in German and Italian cohorts (Supplementary Fig. 2).

**Correlation of SNPs with LOXL1 tissue expression levels.** First, we assessed whether the 14 SNP risk haplotype correlated with tissue expression levels of LOXL1. DNA samples of various post-mortem ocular tissues isolated from 52 individuals with manifest PEX syndrome/glaucoma and 51 individuals without PEX were used for genotyping the 14 intronic SNPs (together with the two non-synonymous SNPs rs1048661 and rs3825942), while mRNA expression levels of *LOXL1* were analysed by quantitative real time (qRT)-PCR. All 103 samples were stratified by their genotypes (risk, non-risk, heterozygous). The 14 SNP risk haplotype (14sR: GTTCTCGCGTAGCG) in the homozygous state was overrepresented (64.7%) in ocular tissue samples derived from PEX patients compared with those (21.2%) obtained from normal donors without PEX, whereas the non-risk haplotype (14sN: ACCGCAAGCCGCTA) was clearly under-represented in PEX samples (5.8%) compared with control samples (27.6%); heterozygous allele combinations were present in 29.4% of PEX and in 51.1% of control samples (Fig. 3a). The presence of the 14 SNP risk haplotype was invariably associated with the presence of the risk haplotype GG of rs1048661 and rs3825942 (haplotype 4, Supplementary Table 2), whereas the

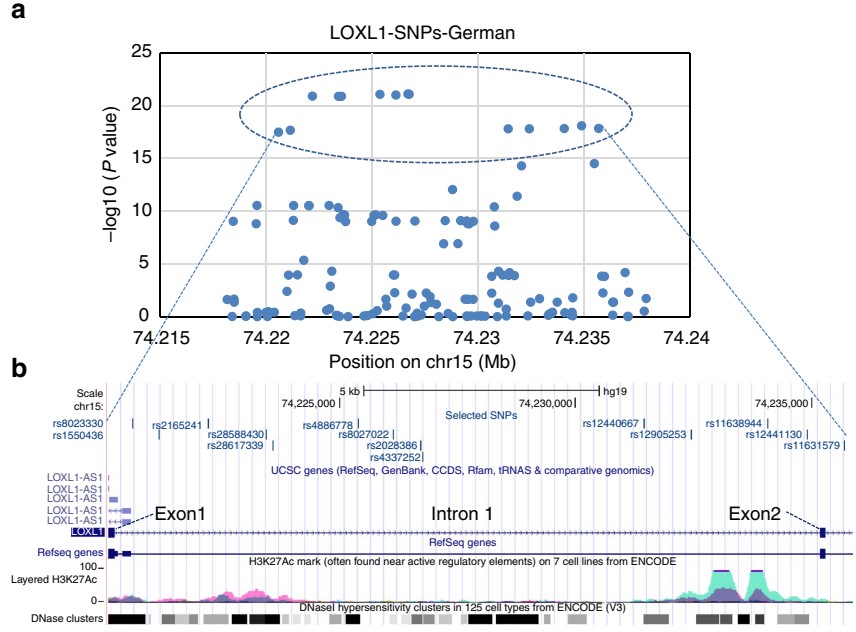

**Figure 2 | Zoom of the regional association plot on the *LOXL1* gene locus based on German data set. (a)**. The zoom of Fig. 1 into the *LOXL1* locus (chr15:74,218,178-74,237,946) shows the 14 selected SNPs with P values $<10^{-15}$ (encircled by dotted line). **(b)** Distribution of the 14 SNPs within the *LOXL1* gene region encompassing exon 1, intron 1, exon 2 and start of intron 2. SNPs ID and their localization are shown using a modified screen shot from the UCSC genome browser (http://genome.ucsc.edu: GRCh37/hg19). ENCODE Regulation Tracks indicate layered histone marks for H3K27Ac (often found near regulatory elements) and DNAse hypersensitivity regions (corresponding to open chromatin indicating active regulatory sites).

**Table 1 | Association analysis for the 14 selected SNPs in three different populations.**

| SNP | Chr | Pos | Ass-all_DE + IT | German (771 Patients/1,350 Controls) | | Italian (421 Patients/1,505 Controls) | | Japanese (1,484 Patients/1,188 Controls) | | |
|---|---|---|---|---|---|---|---|---|---|---|
| | | | | P value_DE | OR (95%CI)_DE | P value_IT | OR (95%CI)_IT | Ass-all_JAP | P value_JAP | OR (95%CI)_JAP |
| rs8023330 | 15 | 74220599 | G | 3.20E − 18 | 2.78 (2.42–3.18) | 3.75E − 29 | 2.93 (2.94–4.14) | A | 2.06E − 35 | 4.73 (3.63–6.16) |
| rs1550436 | 15 | 74221157 | T | 2.09E − 18 | 2.79 (2.43–3.19) | 5.23E − 29 | 2.95 (2.96–4.16) | C | 2.06E − 35 | 4.73 (3.63–6.16) |
| rs2165241 | 15 | 74222202 | T | 1.22E − 21 | 3.02 (2.62–3.47) | 1.54E − 31 | 3.85 (3.22–4.59) | G | 3.95E − 35 | 4.69 (3.60–6.11) |
| rs28588430 | 15 | 74223430 | C | 1.30E − 21 | 3.02 (2.62–3.47) | 1.59E − 31 | 3.85 (3.19–4.49) | G | 1.48E − 35 | 4.75 (3.65–6.18) |
| rs28617339 | 15 | 74223571 | T | 1.27E − 21 | 3.02 (2.62–3.47) | 1.58E − 31 | 3.85 (3.19–4.49) | C | 1.48E − 35 | 4.75 (3.65–6.18) |
| rs4886778 | 15 | 74225388 | C | 8.30E − 22 | 3.1 (2.69–3.57) | 1.54E − 32 | 3.95 (3.28–4.73) | A | 1.53E − 136 | 6.85 (5.73–8.20) |
| rs8027022 | 15 | 74226138 | G | 9.74E − 22 | 3.1 (2.69–3.57) | 2.13E − 32 | 3.95 (3.28–4.73) | A | 2.79E − 136 | 6.84 (5.72–8.18) |
| rs2028386 | 15 | 74226708 | C | 7.62E − 22 | 3.1 (2.69–3.57) | 2.41E − 32 | 3.95 (3.28–4.74) | G | 7.87E − 136 | 6.84 (5.72–8.18) |
| rs4337252 | 15 | 74226765 | G | 8.23E − 22 | 3.1 (2.69–3.57) | 1.38E − 32 | 3.95 (3.25–4.59) | C | 7.87E − 136 | 6.84 (5.72–8.18) |
| rs12440667 | 15 | 74231439 | T | 1.55E − 18 | 2.78 (2.43–3.19) | 1.29E − 29 | 3.55 (2.98–4.22) | C | 2.38E − 35 | 4.71 (3.62–6.13) |
| rs12905253 | 15 | 74232437 | A | 1.55E − 18 | 2.78 (2.43–3.19) | 1.23E − 29 | 3.55 (2.99–4.22) | G | 1.90E − 35 | 4.73 (3.64–6.15) |
| rs11638944 | 15 | 74234082 | G | 1.45E − 18 | 2.78 (2.43–3.19) | 1.16E − 29 | 3.55 (2.99–4.22) | C | 1.90E − 35 | 4.73 (3.64–6.15) |
| rs12441130 | 15 | 74234902 | C | 8.01E − 19 | 2.83 (2.47–3.25) | 5.88E − 31 | 3.65 (3.05–4.36) | T | 5.05E − 137 | 6.89 (5.76–8.24) |
| rs11631579 | 15 | 74235704 | G | 1.41E − 18 | 2.78 (2.43–3.19) | 1.20E − 29 | 3.54 (2.98–4.21) | A | 1.90E − 35 | 4.73 (3.64–6.15) |

Ass-all, associated allele; Chr, chromosome; DE, German; IT, Italian; JAP, Japanese; OR, odds ratio; Pos, position based on UCSC Genome Browser (hg19).
All genotypes were imputed from GWAS data. A logistic regression model adjusted for age and gender was used for the German and Italian cohort, and adjusted for the first six principal components of genetic stratification for the Japanese cohort.

14 SNP non-risk haplotype was associated with haplotypes TG and GA (haplotypes 1 and 5, Supplementary Table 2) of the two non-synonymous variants.

Tissue expression levels of *LOXL1* were found to correlate with the 14 SNP haplotype, with significantly reduced expression levels ($-40$ to 50%, $P < 0.05$) in homozygous risk allele carriers compared to non-risk allele carriers throughout all ocular tissues examined, including iris ($n = 103$), ciliary body ($n = 103$) and lamina cribrosa ($n = 30$) (Fig. 3b–d). Reduced-expression levels of LOXL1 ($-30\%$; $P < 0.001$), indicated by a specific band at 52 kDa corresponding to a processed form of LOXL1, could be also confirmed at the protein level for homozygous carriers of the risk variants ($n = 6$; Fig. 3e, Supplementary Fig. 3). In contrast, there was no correlation between haplotype and tissue expression levels of *LOXL1-AS1* (Supplementary Fig. 4).

**Determination of disease-relevant tissues and cell types.** To analyse the impact of candidate SNPs on transcriptional activity of *LOXL1*, we aimed to determine appropriate disease-relevant cell types, in which the SNPs could exert their regulatory effects. For that purpose, we compared mRNA expression levels of *LOXL1* in intra- and extraocular tissues in different subsets of donors with manifest PEX syndrome and age-matched control subjects carrying the 14 SNP risk and protective haplotypes, respectively. Using qRT-PCR, we observed significantly lower (30–60%) expression levels of *LOXL1* in all tissues of the anterior eye segment, that is, Tenon's capsule ($n = 5$), cornea ($n = 6$), trabecular meshwork ($n = 17$), iris ($n = 32$) and ciliary body ($n = 32$) ($P < 0.001$) (Fig. 4). In posterior segment tissues, reduced expression levels of *LOXL1* were confined to the lamina cribrosa ($n = 20$) of PEX eyes, whereas retina, choroid and sclera samples ($n = 20$ each) showed no differential expression levels between PEX and control subjects. In addition, aortic wall specimens ($n = 5$) derived from PEX patients showed also significantly reduced expression levels of *LOXL1* compared with control specimens ($P < 0.05$) (Fig. 4). These differential expression patterns, which might be explained by tissue-specific regulatory mechanisms, are consistent with previous observations of dysregulated tissue expression of LOXL1 as a hallmark of PEX disease[23–25]. For functional validation of putative regulatory variants in *LOXL1*, we therefore selected human Tenon's capsule fibroblasts (hTCF), corneolimbal epithelial cells (hLEPC), trabecular meshwork cells (hTMC), nonpigmented ciliary epithelial cells (hNPEC), optic nerve head astrocytes (hONHA)

and aortic smooth muscle cells (hASMC) as disease-relevant and 3T3 fibroblasts and HEK293T cells as unrelated cellular models.

**Effect of SNPs on *LOXL1* promoter activity *in vitro*.** To analyse the allele-specific effects of candidate SNPs on transcriptional activity of the human *LOXL1* promoter, 738 bp fragments were constructed containing the haplotypes of all 14 risk (14sR) and non-risk alleles (14sN) and cloned into pGL4.10 luciferase reporter plasmids upstream of a *LOXL1* minimal promoter (nucleotides $-1,438/+1$)[35] (Fig. 5a). Haplotype-specific reporter constructs were transiently transfected into all cell types used. Dual luciferase reporter assays showed a significantly lower reporter activity (35–45%; $P < 0.0001$) for the 14sR haplotype constructs compared with the 14sN constructs in hTCF, hTMC, hNPEC, hONHA and hASMC, but no effect on *LOXL1* promoter activity in hLEPC, HEK293T cells and 3T3 fibroblasts indicating a cell-type specificity of gene regulation (Fig. 5b). Further transfections comparing *LOXL1* promoter activity in hTCF derived from different donors (age < 20 years vs. age > 65 years, presence vs. absence of PEX, German vs. Japanese origin) did not show any statistically significant differences in the decrease of reporter activity by the 14sR/N constructs between subgroups precluding any modulating effect of age, ethnicity or PEX disease in this cell model (Supplementary Fig. 5A).

Next, several deletion constructs of pGL4.10-LOXL1-14sR/N were transfected into hTCF showing that the reduction in *LOXL1* promoter activity (41%; $P < 0.0001$) was achieved by constructs (14sR11–14) including the risk haplotype of the distal four SNPs 11–14. In contrast, constructs including SNPs 1–5 (14sR1–5) or SNPs 6–10 (14sR6–10) failed to differentially affect *LOXL1* promoter activity in hTCF (Fig. 5a,c). Control transfections with empty basic pGL4.10 vector and basal *LOXL1* promoter construct showed only little background luciferase activity and no obvious enhancer activity over core promoter activity (Fig. 5c). Thus, haplotype constructs served as useful indicator of a potential regulatory region including SNPs 11–14 of *LOXL1*.

To ultimately assess a difference in transcriptional activities between individual SNPs included within the effective pGL4.10-LOXL1-14sR/N11-14 constructs, we generated eight distinct reporter plasmids containing each of the four risk or non-risk alleles of SNPs 11–14 in their unaltered genomic context and transiently transfected them into various cell types. Statistically significant allele-dependent effects on reporter activity were observed for SNP 12 and SNP 14 in a cell type-specific manner in

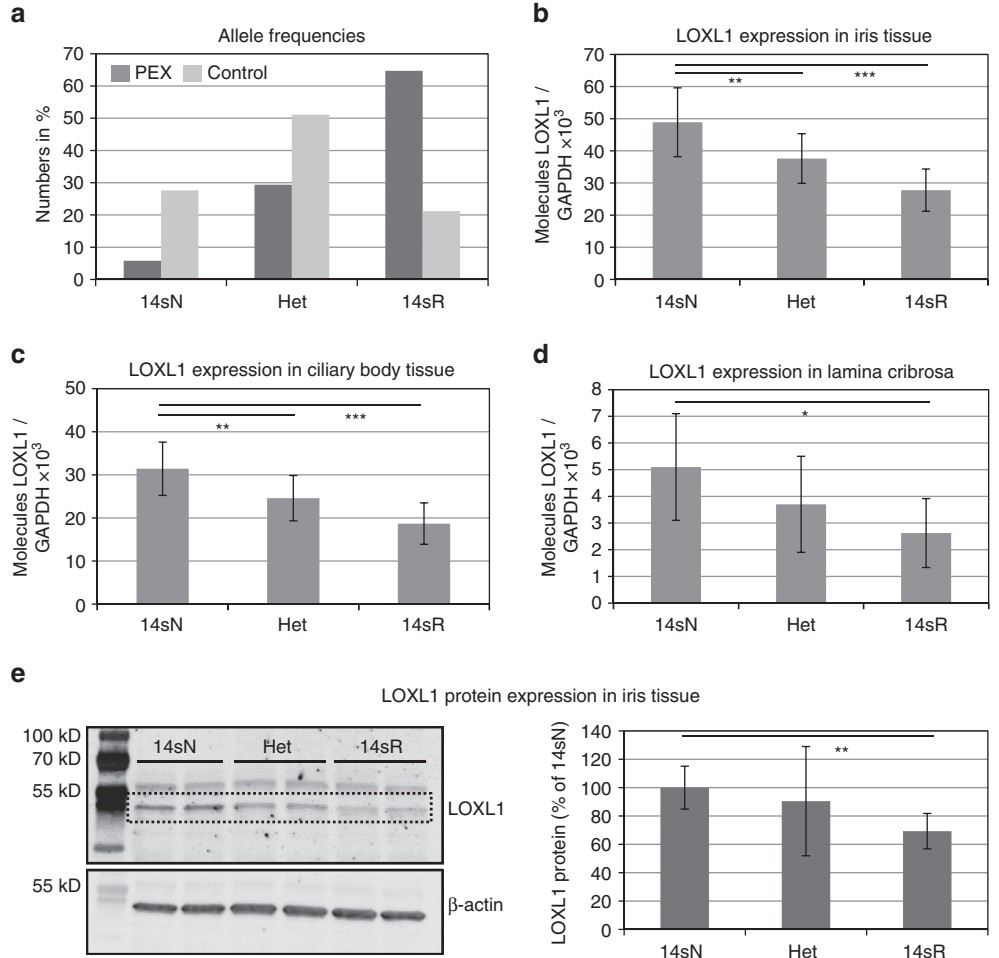

**Figure 3 | Correlation of *LOXL1* risk variants with LOXL1 tissue expression levels.** (**a**) Frequency of the homozygous risk (14sR), non-risk (14sN) and heterozygous (Het) haplotypes formed by the 14 selected SNPs in PEX ($n = 52$) and control ($n = 51$) samples. (**b**) Genotype-correlated expression levels of *LOXL1* mRNA in iris tissue samples ($n = 103$) obtained from PEX ($n = 52$) and control ($n = 51$) patients using real time PCR technology; data are presented as mean values ± s.d. (**c**) Genotype-correlated expression levels of *LOXL1* mRNA in ciliary body tissue ($n = 103$). (**d**) Genotype-correlated expression levels of *LOXL1* mRNA in lamina cribrosa specimens ($n = 30$). (**e**) Western Blot analysis of LOXL1 protein expression in iris specimens ($n = 18$) according to *LOXL1* genotypes; specific bands indicating LOXL1 appear at 52 kDa. Equal loading of samples was verified by immunodetection of ß-actin, and LOXL1 expression levels were normalized to ß-actin expression. Densitometry analysis of band intensities shows mean values ± s.d. of three independent experiments ($n = 6$ samples for each genotype) (the two other Western blots are shown in Supplementary Fig. 3) (*$P < 0.05$; **$P < 0.001$; ***$P < 0.0001$; unpaired two-tailed Student's *t*-test).

hTCF (Fig. 5d), hASMC (Fig. 5e), hTMC, hNPEC and hONHA but not in HEK293T cells (Supplementary Fig. 5B). However, in contrast to the haplotype constructs, the risk alleles increased transcriptional activity compared to non-risk alleles (SNP 12: 35–75%, $P < 0.005$; SNP 14: 25–45%, $P < 0.05$), and the risk sequence of SNP 12 even enhanced transcription up to 120% ($P < 0.005$) from minimal core promoter (Fig. 5d,e). From these findings, which we believe to reflect the true function of these variants in their genomic context better than the engineered haplotypic combinations, we conclude that the risk allele of SNP 12 located at the end of intron 1 of *LOXL1* has enhancer-like function increasing transcriptional activity compared to non-risk alleles in disease-relevant cell types.

**Effect of SNPs on *LOXL1* transcriptional activity *in vivo*.** Since reporter-based assays fail to account for the native chromatin context of the regulatory region, we analysed allele-specific effects of intronic SNPs 11–14 on *LOXL1* transcriptional activity in heterozygous cell lines by allelic expression imbalance assay,

which is considered a robust *in vivo* measure of *cis*-regulatory variants. To discriminate between the two alleles, we used a highly sensitive Taqman SNP Genotyping Assay available for SNP 13 (rs12441130:T > C) serving as a marker SNP for cDNA generated from pre-mRNA transcribed from each chromosomal allele and genomic DNA from the same samples as a control. Pre-mRNA analysis from hTCF ($n = 15$) heterozygous at all four SNPs 11–14 showed distinct allelic imbalance compared to genomic DNA with an allelic shift towards the risk allele C (Fig. 6a). The relative abundance of the risk allele C was threefold increased over that of the non-risk allele T ($P < 0.0001$).

These changes in relative allelic rates of transcription could be confirmed by RNA polymerase II (Pol II)-ChIP, which was performed with cross-linked chromatin from heterozygous hTCF ($n = 2$) using antibodies against Pol II, histone H3 and acetylated histone H3K26Ac (positive controls), and non-immune rabbit IgG (negative control), and allele-specific Taqman SNP Genotyping Assay for rs12441130 (SNP 13) (Fig. 6b). Almost twofold increased relative amounts of DNA containing the risk alleles were precipitated with the Pol II antibody (Fig. 6b), suggesting

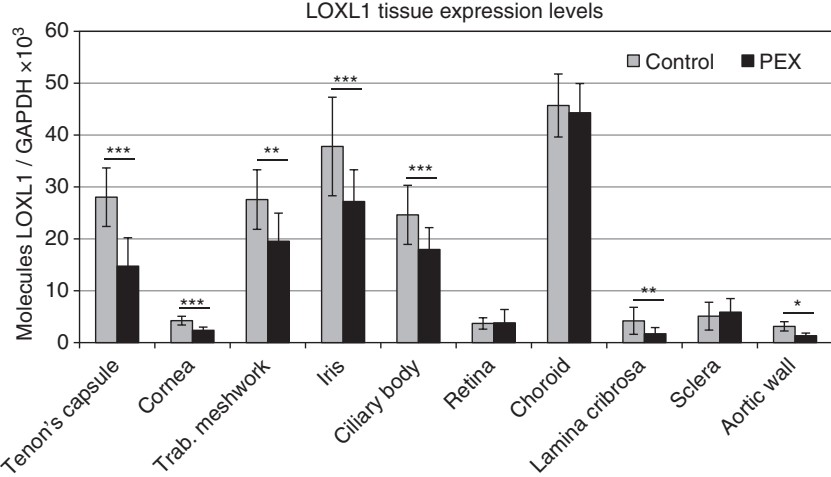

**Figure 4 | Determination of pathophysiologically relevant tissues and cell types.** Expression of *LOXL1* mRNA in ocular and extraocular tissues derived from normal human donors (control) and donors with manifest PEX syndrome using real-time PCR technology. The expression levels were normalized relative to GAPDH. Expression levels were significantly reduced in Tenon's capsule ($n=5$), cornea ($n=6$), trabecular meshwork ($n=17$), iris ($n=32$), ciliary body ($n=32$), lamina cribrosa ($n=20$) and aortic wall ($n=5$) from PEX patients compared to controls, and were not different in retina, choroid and sclera ($n=20$ each); (data represent mean values ± s.d.; *$P<0.05$, **$P<0.001$, ***$P<0.0001$; unpaired two-tailed Student's *t*-test).

that the transcriptional activity at the risk sequence is increased over that of the non-risk sequence. Antibodies against histone H3 did not show any difference between both alleles. Fidelity of the ChIP assay was confirmed by PCR of immunoprecipitated complexes and by qPCR showing sixfold enrichment of Pol II over IgG control (Fig. 6b).

In view of the fact that the risk haplotype correlated with reduced total *LOXL1* mRNA expression levels in post-mortem tissue samples (Fig. 3b–d), as a next step we examined whether increased transcriptional activity at the risk sequence is associated with post-transcriptional regulation of steady-state gene expression levels by alternative pre-mRNA splicing. ENSEMBL database indicated the presence of an alternative transcript (*LOXL1*-002: ENST00000566011; Havana project, version 1: transcript OTTHUMT00000419872.1), which is characterized by selective inclusion of an additional exon (E1A) between exon 1 and exon 2 of the *LOXL1* gene introducing a premature termination codon in exon 2 via frameshift (Fig. 7a). The premature termination codon is localized 100 nucleotides upstream of the exon 2–exon 3 junction thus fulfilling the requirements ($>50$ nucleotides upstream of an exon/exon junction complex) for a NMD target[36]. Transcript levels of *LOXL1-E1A* were determined by qPCR using exon 1-specific forward and E1A-specific reverse primers (Supplementary Table 3.2) and identity of qPCR fragments was confirmed by sequence analysis (Eurofins Genomics). Treatment of hTCF with the protein synthesis inhibitor puromycin, which is known to selectively stabilize NMD transcripts[36], increased the level of *LOXL1-E1A* fivefold ($P<0.001$), whereas levels of wild-type *LOXL1* mRNA were decreased fourfold ($P<0.0001$) (Fig. 7b). In contrast, levels of *LOXL1-AS1* were not significantly changed by puromycin. RT-PCR using primers spanning E1A (Supplementary Table 3.3) produced a prominent band for wild-type *LOXL1* at 147 bp and a weaker band for *LOXL1-E1A* at 331 bp after treatment with puromycin in hTCF cell lines (Fig. 7b). Genotype–phenotype correlations showed that hTCF cell lines homozygous for the risk allele C of the marker SNP 13 express significantly higher constitutive levels of *LOXL1-E1A* (1.75-fold; $P<0.05$) and significantly lower levels of wild-type *LOXL1* mRNA ($-1.65$-fold; $P<0.001$) than cell lines homozygous for the protective allele T, whereas levels of *LOXL1-AS1* did not correlate with the genotype of cells (Fig. 7c).

To ultimately confirm that *LOXL1-E1A* is a direct target of NMD, we inhibited NMD through transient knockdown of UPF1 (Up Frame Shift Protein 1), a central component of the NMD pathway[36], leading to a significant increase in *LOXL1-E1A* in hTCF (Fig. 7d).

From these experiments we conclude that alternative splicing of *LOXL1* pre-mRNA coupled with NMD is significantly influenced by the genotype of SNPs 11–14 located in close proximity to the splice site (Fig. 7a). Higher rates of NMD lead to reduced steady-state levels of *LOXL1* in risk allele carriers compared to non-risk allele carriers. This common mechanism of post-transcriptional regulation of gene expression[36] may explain downregulation of total *LOXL1* expression levels in cells and tissues of risk allele carriers despite of increased transcriptional activity at the risk sequence.

**Effect of SNPs on transcription factor binding affinity.** Since transcription factors have been found as primary mediators of sequence-specific regulation of gene expression in a context-dependent manner, we performed electrophoretic mobility shift assays (EMSAs) to determine allele-specific differences in transcription factor–DNA interactions. Using biotinylated 228–292 bp DNA fragments containing the risk and non-risk alleles of individual SNPs 11–14 and nuclear extracts from disease-relevant cell types (hTMC, hNPEC, hTCF), all probes yielded one or two specific shifted bands, which could be competitively inhibited by unlabelled oligonucleotides up to 100% (Supplementary Fig. 6). Quantitative analysis of the shifted bands relative to the unshifted bands from 5 independent experiments revealed that the risk sequences of rs12905253 (SNP 11) and rs12441130 (SNP 13) showed greater binding affinity to nuclear proteins (35–57%; $P<0.05$) compared to the non-risk sequences. In contrast, the risk sequences of rs11638944 (SNP 12) and rs11631579 (SNP 14) showed lower binding efficiency ($-25\%$) than the non-risk sequences. These allelic differences were statistically most significant ($P<0.005$) for SNP 12-containing probes, which also showed the greatest binding affinity to nuclear proteins of all probes (Supplementary Fig. 6). No allele-specific differences were observed with nuclear extracts from hLEPC, HEK293T and 3T3 cells (Supplementary Fig. 7).

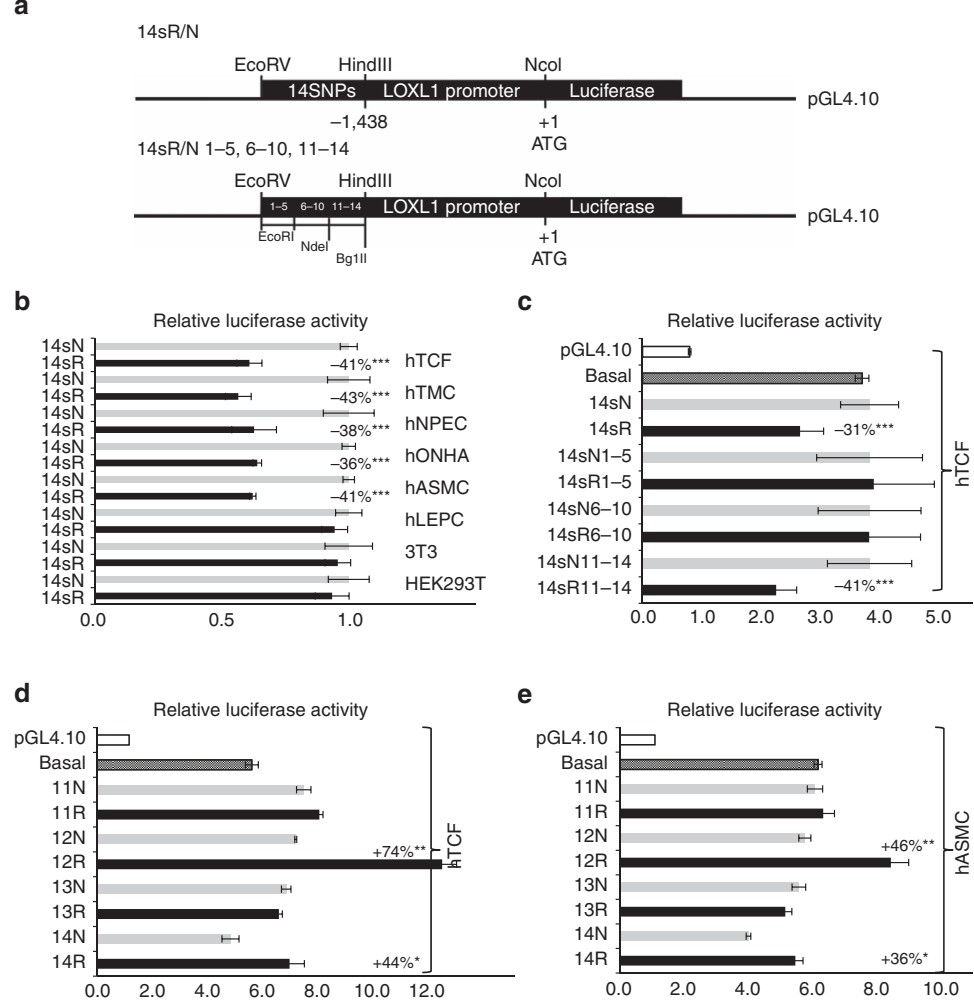

**Figure 5 | Effects of risk variants on *LOXL1* promoter transcriptional activity *in vitro*.** (**a**) Reporter constructs used in transfection experiments outlining the 14 S/R constructs comprising all 14 SNPs flanked by 51 bp of genomic DNA sequences each (top) and three deletion constructs comprising SNPs 1–5, SNPs 6–10 and SNPs 11–14, respectively (bottom). (**b**) Dual luciferase reporter assays demonstrating regulatory activity of 14 SNP risk (14sR) and non-risk (14sN) haplotypes in human Tenon's capsule fibroblasts (hTCF), trabecular meshwork cells (hTMC), nonpigmented ciliary epithelial cells (hNPEC), optic nerve head astrocytes (hONHA), aortic smooth muscle cells (hASMC), limbal epithelial cells (hLEPC), 3T3 fibroblasts and HEK293T cells. Results are expressed as the ratio of Firefly luciferase to Renilla luciferase; the transcriptional activity of the non-risk sequence was set at 100%. (**c**) Regulatory activity of three deletion constructs (SNPs 1–5, 6–10 and 11–14) in hTCF compared with the basal *LOXL1* promoter activity; the transcriptional activity of the empty pGL4.10 vector was set at 100%. (**d,e**) Activity assays using reporter plasmids containing each of the four risk or non-risk alleles of individual SNPs 11, 12, 13 and 14 compared with the basal *LOXL1* promoter activity in hTCF (**d**) and hASMC (**e**); the transcriptional activity of the empty pGL4.10 vector was set at 100% (data represent mean values ± s.d. of five independent experiments; *P<0.05; **P<0.005; ***P<0.0001; unpaired two-tailed Student's t-test).

*In silico* analysis using rSNP-MAPPER software[37] identified several putative transcription factor binding sites potentially affected by the sequence variant differences. Among sites with high predicted difference for an allelic effect on binding were: LXRα:RXRα, myogenin/NF-1, Roaz, ZID, NR2F1, NFκB, PPARα:RXRα and HNF-4α in the region of rs11638944 (SNP 12) (Supplementary Table 4, Supplementary Fig. 8A). Supershift assays using DNA probe S12R and antibodies against candidate transcription factors did not show any specific binding of antibodies against LXRα, myogenin, NF-1, Roaz, NR2F1, NFκBp50, NFκBp65 (RELA), PPARα, PPARγ and HNF-4α, whereas antibodies against RXRα (retinoid X receptor alpha) and ZID (zinc finger protein with interaction domain) disrupted the protein–DNA complexes in EMSA experiments and produced supershifted bands (Fig. 8a). Because mRNA expression of lysyl oxidases is known to be regulated by retinoic acid[38], we further focused on the nuclear retinoic acid receptor RXRα, which mediates retinoic acid-induced gene activation. Comparison of risk and non-risk sequences of all four SNPs in supershift experiments with anti-RXRα antibody showed increased supershifted band densities for the risk sequences of SNP 11 and SNP 13 by 25–150%; differences were, however, statistically not significant. In contrast, binding affinity of RXRα antibody to the risk sequences of SNP 12 (−26%; P<0.001) and SNP 14 (−17%; NS) was decreased compared to the non-risk sequences (Fig. 8b, Supplementary Fig. 9), although predicted binding scores were higher for the risk than for the non-risk sequences of SNP 12 (Supplementary Table 4, Supplementary Fig. 8B).

In view of this discrepancy, specific binding of both recombinant RXRα protein fragment containing the DNA-binding domain (in a concentration-dependent manner) and full-length RXRα protein was used as positive control and resulted in shifted bands corresponding to their respective molecular sizes, whereas recombinant RXRα fragment containing the ligand-binding

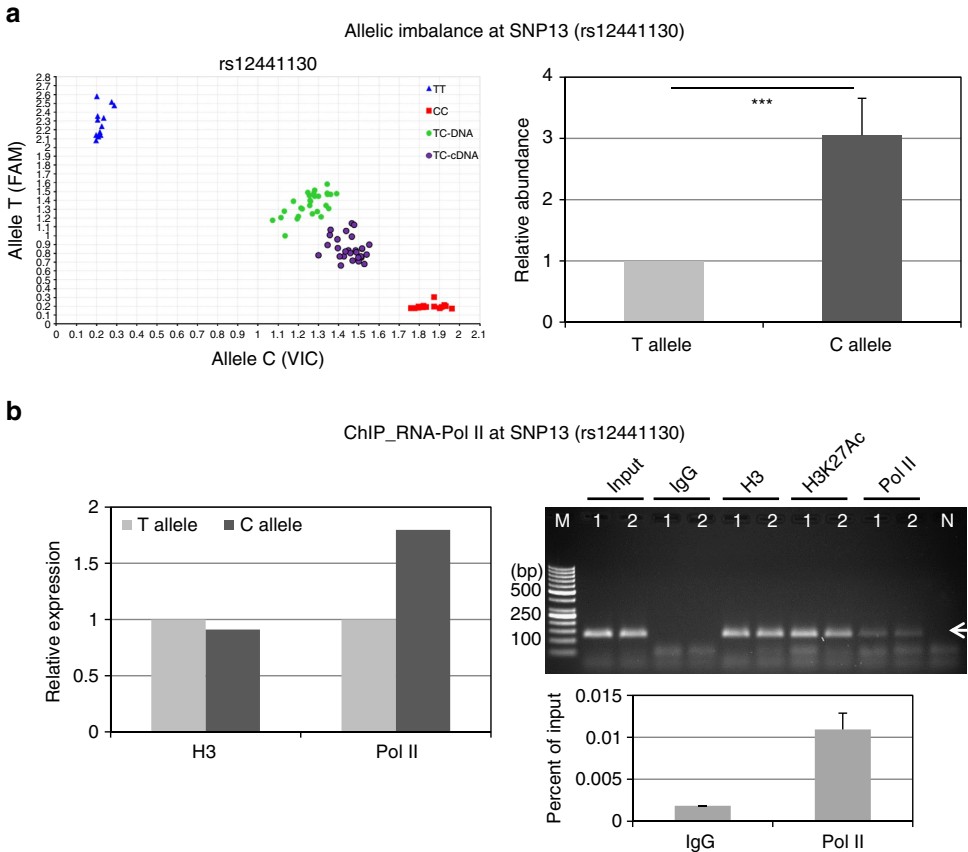

**Figure 6 | Effects of risk variants on *LOXL1* transcriptional activity *in vivo*.** (**a**) Scatter plot of TaqMan-based allelic discrimination of the *LOXL1* SNP 13 (rs12441130). The genotypes of hTCF homoyzgous for the risk (C) or non-risk (T) alleles are shown in relation to genomic DNA and pre-mRNA containing cDNA of heterozygous hTCF cell lines (*n* = 15). Relative abundance of risk allele C over non-risk allele T in heterozygous hTCF cell lines (*n* = 15); expression level of T allele was set at 100%. (**b**) ChIP assay for RNA polymerase II (Pol II) binding at rs12441130 (SNP13)-containing region of *LOXL1* in heterozygous hTCF cell lines (*n* = 2) using antibodies against Pol II, histone H3 and acetylated histone H3K27Ac (positive controls), and non-immune IgG (negative control); input represents total chromatin applied for immunoprecipitation. Allele-specific ChIP-qPCR analysis for Pol II chromatin binding and histone H3 is shown (left); expression levels of the non-risk allele T were set at 100%. DNA isolated from immunoprecipitated complexes was analysed on 2% agarose gel (top right) and by qPCR (bottom right) with primers specific for the SNP13 region producing a 121 bp PCR fragment (arrow). Data are expressed as per cent of input (Lane 1: hTCF 1, lane 2: hTCF 2, lane M: DNA marker, lane N: primer control without chromatin).

domain was not effective (Supplementary Fig. 10A). A biotinylated control oligonucleotide containing a RXR consensus binding site was further used as positive control in EMSAs using hTCF, hTMC and hNPEC nuclear extracts and resulted in shifted bands which could be competitively inhibited by unlabelled probes (Supplementary Fig. 10B). Moreover, RXRα binding to SNP 12 region *in vivo* was verified by RXRα-ChIP, which was performed with cross-linked chromatin from hTCF (*n* = 2) using two different antibodies against RXRα, H3 (positive control) and non-immune IgG (negative control). Fidelity of the ChIP assay was confirmed by PCR of immunoprecipitated complexes and by qPCR showing fourfold enrichment of RXRα over IgG control (Fig. 9a). Because a specific Taqman Genotyping Assay for SNP 12 was not commercially available to assess allele-specific differences in RXRα binding, we performed luciferase assays using both risk and non-risk genotypes of SNP 12 without or with co-transfection of RXRα-specific siRNA in hTCF. These experiments showed that the allele-specific differences in transcriptional activity were abolished by knockdown of RXRα (Fig. 9b), providing additional evidence for an allele-specific binding of RXRα and a role for this transcription factor in *LOXL1* gene regulation.

RXRα was found to be expressed in nuclear fractions of all disease-relevant cell types including hTCF, hTMC, hNPEC and

hASMC (Fig. 9c). siRNA-mediated knockdown of RXRα in hTCF resulted in a significant upregulation (40%; *P* < 0.005) of *LOXL1* mRNA expression compared to mock-transfected control cells (Fig. 9d). Stimulation with all *trans*-retinoic acid (2 μM for 24 h) significantly increased *RXRα* expression but reduced *LOXL1* expression in disease-relevant cell types, such as hTCF and hASMC, compared with non-treated controls (Fig. 9e). These data provide direct evidence that RXRα influences *LOXL1* expression by functioning as a transcriptional repressor.

Sequence inspection of the DNA probes using rSNP-MAPPER software predicted binding motifs for RXRα:LXRα and RXRα:P-PARα heterodimers in the region of SNP 12, which overlapped the SNP position (Supplementary Fig. 8B). However, antibodies against known heterodimeric partners for RXRα, that is, RARα, LXRα/β, PPARα/γ, VDR, TRα/β and ERα, did not result in supershifted bands (Fig. 8a, Supplementary Fig. 11) suggesting either homodimeric RXRα binding or another not yet identified heterodimeric binding partner. Nevertheless, the present findings identified RXRα as candidate transcription factor showing significantly reduced binding to the risk sequence of rs11638944, which may be important for retinoic acid sensitivity of this regulatory region and for the suppressive effect of retinoic acid on *LOXL1* expression.

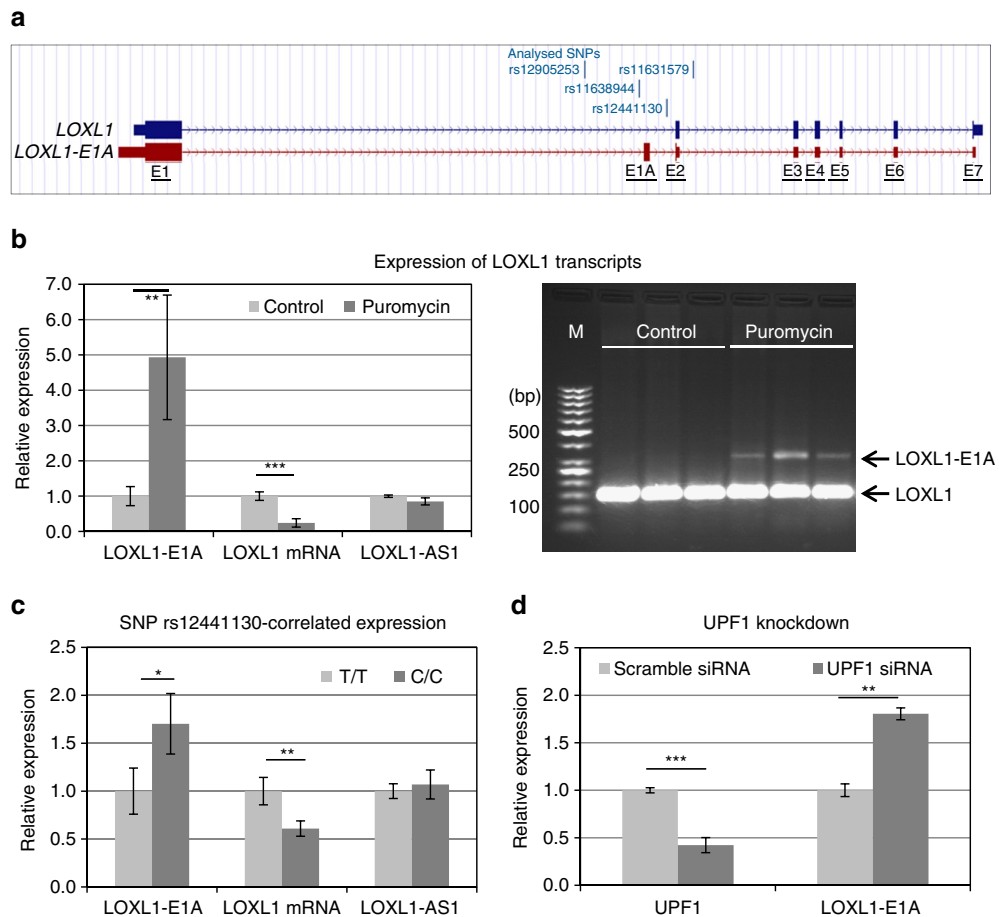

**Figure 7 | Effects of risk variants on LOXL1 alternative splicing.** (**a**) The two *LOXL1* transcripts (*LOXL1, LOXL1-E1A*) seen in the alignment with position of SNPs 11–14; exons are indicated by numbers (E1–E7), the additional exon included in the alternative transcript is indicated as E1A; the location of stop codons is marked by vertical lines. (**b**) Expression levels of transcripts *LOXL1-E1A*, *LOXL1* and *LOXL1-AS1* in hTCF cell lines (*n* = 16) without (control) or after treatment with puromycin (200 ng ml$^{-1}$ for 10 h) using real-time PCR technology (left); hTCF were either homozygous for the risk (*n* = 8) or non-risk (*n* = 8) alleles for SNPs 1–14. PCR from *LOXL1* and *LOXL1-E1A* transcripts without and after puromycin treatment of hTCF (*n* = 3) using primer pairs spanning exon 1A (right; lane M: DNA marker). (**c**) SNP rs12441130-correlated expression levels of transcripts *LOXL1-E1A*, *LOXL1* and *LOXL1-AS1* in hTCF cell lines homozygous for the risk alleles (C/C) (*n* = 8) or non-risk alleles (T/T) (*n* = 8) using real-time PCR technology. (**d**) Real-time PCR analysis of *UPF1* and *LOXL1-E1A* in hTCF cell lines (*n* = 4) transfected with UPF1-specific siRNA or scrambled control siRNA; expression levels were normalized relative to GAPDH (data represent mean values ± s.d.; *$P$ < 0.05; **$P$ < 0.001; ***$P$ < 0.0001; unpaired two-tailed Student's *t*-test).

## Discussion

Genetic and pathogenetic analyses have confirmed a highly significant and direct role for LOXL1 in the pathophysiology of PEX syndrome and its associated ocular and systemic complications. Dysregulated tissue expression of LOXL1 has been shown to be a hallmark of PEX syndrome, contributing to disease development and predisposing to ocular and systemic complications[23–26]. Reduced tissue expression levels of LOXL1 have been related to pronounced structural elastotic and biomechanical alterations of elastin-rich connective tissues, such as the lamina cribrosa and blood vessel walls, thereby increasing the risk for pressure-induced optic nerve damage and cardiovascular complications[24,33]. Accordingly, our group and others have described a significant association between PEX and systemic disorders of elastic tissues, that is, aortic aneurysms, coronary artery ectasia, renal artery stenosis and pelvic organ prolapse[6–8,39–42]. Changes in *LOXL1* gene expression have been observed in many other conditions, including aging[43,44] and disorders involving weakening of elastic connective tissues[45–47].

Although the discovery of *LOXL1* as the principal genetic risk factor for PEX dates back to 2007 (ref. 11) and has been

subsequently replicated by multiple studies in various geographical populations[12–16], it is currently not known how associated variants contribute to disease. So far, no single causal variant has been identified. In fact, the precise analysis of the *LOXL1* locus is still at a rudimentary stage, and the translation of genetic findings into molecular mechanisms remains a hitherto unmet need and challenge. Here, we provide experimental evidence that a common risk variant, located in a genomic region of intron 1 of *LOXL1* with regulatory potential, exerts a *cis*-acting effect on *LOXL1* expression, mediated by differential transcription factor binding and differential alternative pre-mRNA splicing in a cell type-specific manner. We show that increased transcriptional activity at the risk sequence is associated with reduced binding of RXRα and with increased levels of an alternatively spliced *LOXL1* transcript sensitive to NMD, possibly by allele-specific interaction with splicing factors, and, seemingly contradictory, with decreased levels of normal *LOXL1* mRNA in cells and tissues of risk allele carriers.

However, significant inconsistencies in pre-mRNA and mRNA expression levels have been described for many genes and indicate a post-transcriptional mode of regulation frequently

involving the NMD pathway[36,48]. The NMD pathway is a common translation-dependent mRNA degradation pathway that controls levels of alternatively spliced unproductive mRNA

variants to regulate gene expression. In the case of constitutive unproductive splicing, which may be influenced by sequence variation close to splice sites, the combined effect of alternative

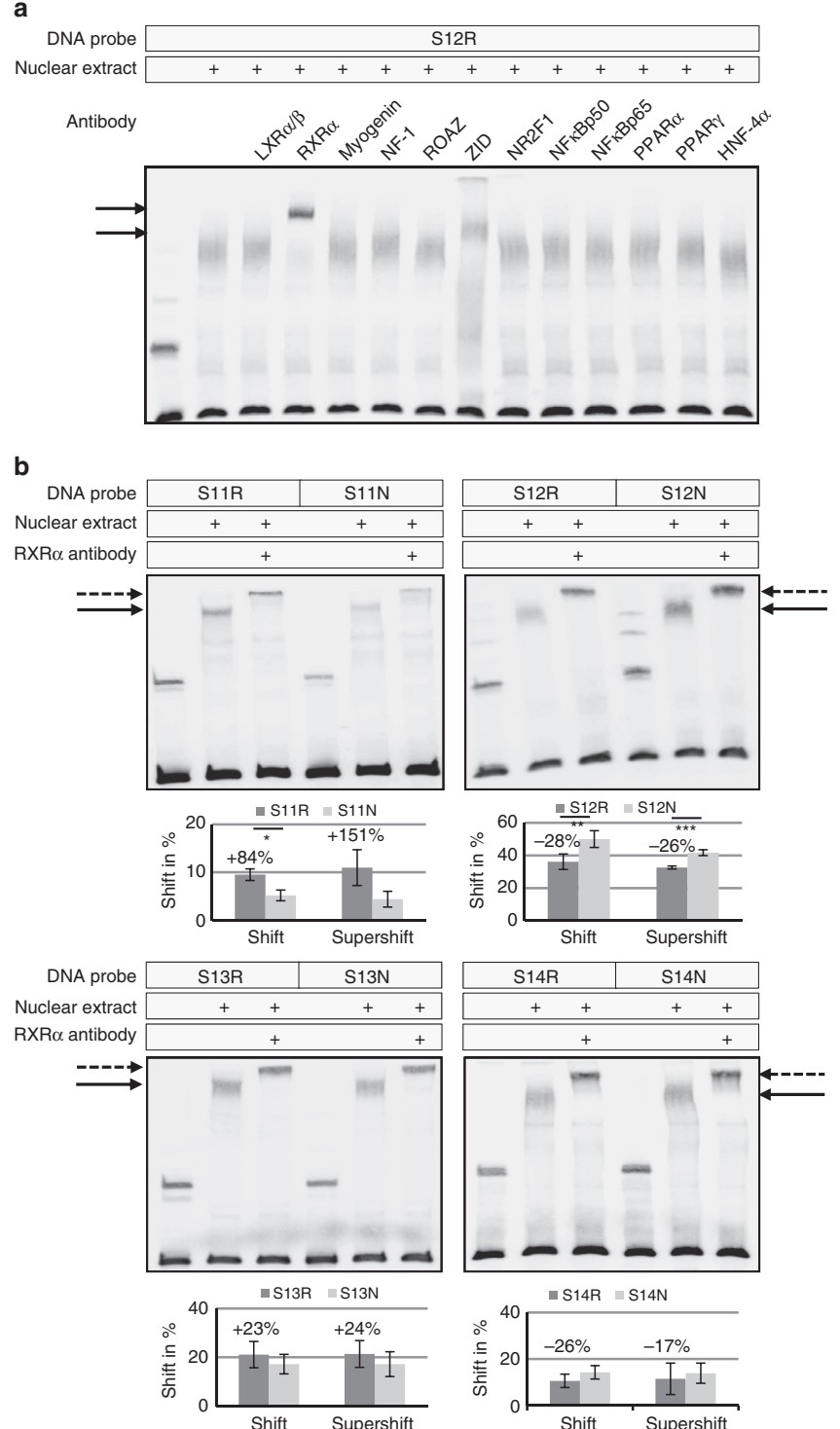

**Figure 8 | Allele-specific transcription factor binding.** (**a**) Supershift assays with DNA fragments containing the risk sequence of rs11638944 (S12R), nuclear extracts from human trabecular meshwork cells, and specific antibodies against candidate transcription factors liver X receptor (LXR)α/ß, retinoid X receptor (RXR)α, myogenin, nuclear factor (NF)-1, Roaz, zinc finger protein with interaction domain (ZID), nuclear receptor subfamiliy 2 group F member 1 (NR2F1), nuclear factor (NF)κBp50, NFκBp65, peroxisome proliferator-activated receptor (PPAR)α, PPARγ, and hepatocyte nuclear factor 4 (HNF-4)α showed supershifted bands (arrows) with RXRα and ZID antibodies. (**b**) Supershift assays with a specific antibody against RXRα disrupted the DNA–protein complexes (solid arrows) to produce distinct supershifted bands (dotted arrows) in a differential manner between DNA fragments containing the risk (R) alleles and fragments containing the non-risk (N) alleles. Quantitative analyses of the (super)shifted bands relative to the unshifted bands show mean values ± s.d. of five independent experiments (*$P < 0.05$; **$P < 0.005$; ***$P < 0.001$; unpaired two-tailed Student's $t$-test).

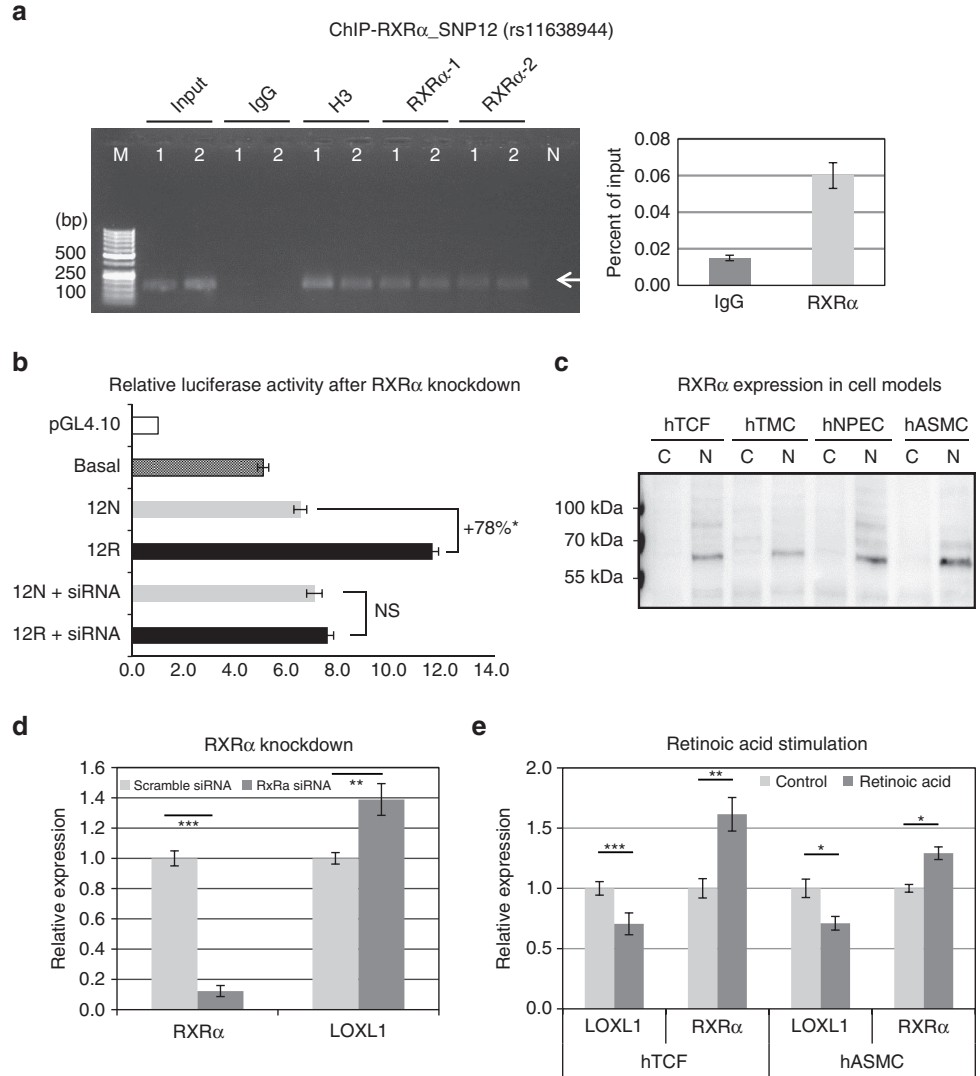

**Figure 9 | RXRα influences _LOXL1_ expression by functioning as a transcriptional repressor.** (**a**) ChIP assay for RXRα binding at rs11638944 (SNP12)-containing region of _LOXL1_ in heterozygous hTCF cell lines ($n = 2$) using two different antibodies against RXRα (1: D-20, 2: F-1), histone H3 (positive control) and non-immune IgG (negative control); input represents total chromatin applied for immunoprecipitation. DNA isolated from immunoprecipitated complexes was analysed on 2% agarose gel (left) and by qPCR (right) with primers specific for the SNP12 region producing a 148 bp PCR fragment (arrow). Data are expressed as per cent of input (Lane 1: hTCF 1, lane 2: hTCF 2, lane M: DNA marker, lane N: primer control without chromatin). (**b**) Dual luciferase reporter assays using reporter plasmids containing risk (12R) and non-risk (12N) genotypes of SNP 12 alone or co-transfected with RXRα-specific siRNA in human Tenon's capsule fibroblasts (hTCF) compared with the basal _LOXL1_ promoter activity. Results are expressed as the ratio of Firefly luciferase to Renilla luciferase; the transcriptional activity of the empty pGL4.10 vector was set at 100% (data represent mean values ± s.d. of three independent experiments; *$P < 0.005$; unpaired two-tailed Student's _t_-test). (**c**) Western blot analysis of RXRα expression in cytoplasmic and nuclear protein fractions of human Tenon's capsule fibroblasts (hTCF), trabecular meshwork cells (hTMC), nonpigmented ciliary epithelial cells (hNPEC) and aortic smooth muscle cells (hASMC) at ~60 kDa (C, cytoplasmic fraction; N nuclear fraction). (**d**) Real-time PCR analysis of _RXRα_ and _LOXL1_ mRNA in hTCF cell lines ($n = 4$) transfected with RXRα-specific siRNA or scrambled control siRNA; expression levels were normalized relative to GAPDH. (**e**) Real-time PCR analysis of _RXRα_ and _LOXL1_ mRNA without and with stimulation by 2 μM all _trans_-retinoic acid for 48 h in hTCF and hASMC; expression levels were normalized relative to GAPDH (data represent mean values ± s.d.; *$P < 0.05$, **$P < 0.001$, ***$P < 0.0001$; unpaired two-tailed Student's _t_-test).

splicing coupled to NMD reduces total message abundance by a more or less constant factor, which has been also observed in cells and tissues of risk allele carriers in the present study. Moreover, coupling of alternative splicing and NMD has been reported as a general mode of controlling gene expression in a dynamic, developmental stage- and cell type-specific manner, promoting cellular adaptation in response to a variety of cellular stresses[49,50]. Preliminary data have shown that _LOXL1-E1A_ activity is modulated by distinct forms of stressors and metabolites, including oxidative stress, to regulate _LOXL1_ gene expression.

Thus, we have identified allele-specific effects on _LOXL1_ gene regulation at the transcriptional (RXRα binding), co-transcriptional (pre-mRNA splicing) and post-transcriptional (mRNA decay) level, together resulting in a downregulation (40–50%) of _LOXL1_ gene expression levels in cells and tissues of risk allele carriers. The findings are further supported by known functions of RXRα generally behaving as constitutive repressor of transcription in the absence of ligand[51] through interaction with a diverse repertoire of canonical and non-canonical binding elements[52].

Although high LD in the region makes it difficult to distinguish the polymorphism(s) driving the causal association, rs11638944:C > G (SNP 12) showed most significant enhancer-like activities and allelic effects on transcription factor binding, suggesting that rs11638944 functions as primary *cis*-acting expression quantitative trait locus (eQTL) of *LOXL1*. The SNP is located in a region of intron 1 with enhancer-like chromatin features, such as specific histone modifications marking active enhancers (H3K27Ac), DNase I hypersensitivity sites and transcription factor binding sites (Supplementary Fig. 8A), which are compatible with predictions of functional regulatory elements[53]. Further confirmation arises from eQTL data sets, such as the BROAD Institute's GTEx portal (www.gtexportal.org/), revealing a significant eQTL for this SNP in pituitary tissue. These findings are also in line with current concepts of regulatory variants affecting gene expression as major contributors to common disease risk[34]. Disease-associated variants in noncoding regions, particularly those that map to enhancer elements specific to disease-relevant cell types, have been estimated to explain a greater proportion of the heritability for some disorders than variants in coding regions[54].

There are, however, some limitations of the study.

1. The PEX-associated risk alleles were identical in two European populations, but completely reversed in Japanese populations (Table 1), which seems to be a principle for all *LOXL1* variants identified so far[21,55–58]. Allelic reversal in Japanese patients may be explained by allelic heterogeneity, the existence of different SNPs tagging the same not yet identified causal variant in all populations, or multi-locus effects implying a change in LD pattern between the two markers in two populations of different ancestry[59]. Such flip–flop associations, probably caused by multi-locus effects, have already been reported for other complex diseases including Alzheimer's disease and autism as well as obesity[60,61]. SNP function may be also context-dependent and the cause of the disparity between European and Japanese populations may be related to a combination of genetic architecture and environmental risk factors causing a similar phenotype. Although this enigma cannot be solved by the present study, potential regulatory differences at the DNA level as well as potential differences in *LOXL1* expression quantitative phenotypes and their relation to variations in PEX prevalence between European and Japanese populations will be the subject of future studies.

2. Other potential mechanisms that are critical to regulation of gene expression and may be modulated by genetic variation, such as epigenetic modifications, disruption of chromatin interactions, action of noncoding RNAs, and mRNA processing and stability have not been investigated in this analysis. DNA hypermethylation of the *LOXL1* promoter has been shown to occur in PEX patients[26] and reported to downregulate *LOXL1* expression in skin fibroblasts in a case with cutis laxa[35] and in human bladder cancer cells[62]. A functional SNP, rs45008784A > C, in the promoter region of *LOXL1* impacting gene expression has been identified, but was not associated with disease (pelvic organ prolapse) and did not reveal any differential transcription factor binding[63]. Hauser *et al.*[22] have identified a region of peak association at the *LOXL1* exon 1–intron 1 junction, which contains a promoter of a long non-coding antisense RNA, *LOXL1-AS1*, which is modulated by a haplotype of three functional risk variants associated with PEX in a South African population. However, this region does not enhance *LOXL1* promoter activity *in vitro* and has not been functionally linked to PEX pathophysiology *in vivo*, although the authors suggested that dysregulated expression of *LOXL1-AS1* plays a role in PEX pathogenesis, particularly in cellular response to stresses.

Despite the limitations mentioned above, we were able to establish a functional link between *LOXL1* genetic variation and regulation of the expression of *LOXL1* through identification of an eQTL in intron 1 explaining part of the genetic variance of the *LOXL1* expression phenotype. The present findings also highlight the importance of context-specificity in resolving the mechanisms of action of regulatory variants. We further suggest molecular mechanisms by which sequence variation modulates transcriptional output on two levels of gene expression regulation, that is, differential transcription factor binding and alternative pre-mRNA splicing, and identified the transcription factor RXRα and regulated coupling of alternative splicing and NMD as a mechanism involved in *LOXL1* gene regulation; their precise functional roles and mechanisms of action will be scrutinized in future studies. Since we were not able to untangle the flipped allele phenomenon observed between Caucasian and Asian populations, targeted resequencing of the risk haplotype in both populations and further context-specific research is required to determine how *LOXL1* variants actually contribute to PEX pathophysiology. Ultimately, the functional translation of genetic information to the tissue level and a detailed understanding of *LOXL1* gene regulation will guide improved risk prediction and future treatment strategies for a common disease and its potentially sight- and life-threatening complications.

## Methods

**Studied groups and study approval.** This study utilized three case–control data sets from two European (German, Italian) and one Asian (Japanese) cohort. Ethics approval for this study was obtained from the Institutional Review Boards of the Medical Facultiy of the University of Erlangen-Nürnberg (Germany), the Medical Faculty of the University of Siena (Italy), the Monfalcone Hospital (Italy), Hayashi Eye Hospital (Fukuoka), Mizoguchi Eye Hospital (Nagasaki), Oita University Faculty of Medicine, Kyoto University Graduate School of Medicine, Ideta Eye Hospital (Kumamoto), the Sing Health Centralised Institutional Review Board Singapore as well as three sites in Miyazaki, that is, Shinjo Eye Clinic, Miyata Eye Hospital and Ozaki Eye Hospital. All participants had given written informed consent, and investigations were performed in accordance with the principles of the Declaration of Helsinki. All patients with PEX syndrome showed manifest PEX material deposits on anterior segment structures in mydriasis on slitlamp biomicroscopy. Secondary open-angle glaucoma due to PEX was defined, if elevated intraocular pressure, characteristic visual field defects and characteristic optic disc atrophy were observed in the presence of an open chamber angle and typical PEX deposits. Healthy population-based controls were recruited from the same geographic areas as the PEX patients[64–67].

To investigate the association between PEX and SNPs in chromosomal region [chr15: 74,218,178-74,237,946 (hg19)] encompassing exon 1, intron 1, exon 2 and part of intron 2 of *LOXL1* (19 kb), we used SNP array-derived data (Affymetrix Genome-Wide Human SNP Array 6.0) of 771 German patients with PEX (PEX syndrome: 309; PEX glaucoma: 462) and 1,350 healthy individuals from a population-based control cohort (KORA). These patients and controls formed the discovery part of our study. To replicate association results of selected SNPs, we used SNP array-derived data (Illumina OMNI Chip) of 473 Italian PEX patients (421 with age and gender information) and 1,545 healthy individuals (1,505 with age and gender information) from an Italian population-based control cohort (Hypergene) as well as 1,484 Japanese PEX patients and 1,188 healthy individuals from a Japanese population-based control cohort.

**Genotyping and association analysis.** Genotyping of German patients and control subjects was performed on the Affymetrix Genome-Wide Human SNP Array 6.0. Italian patients were genotyped on the Illumina OmniExpress Bead Array-12 v.1.1, Italian control subjects on the Illumina 1M Duo. Stringent quality checks for the German data set were performed on a per-SNP and per-sample basis, with removal of SNPs showing a genotyping success rate of <95% or a minor allele frequency of <1% as well as deviation from Hardy–Weinberg equilibrium ($P < 1 \times 10^{-3}$ for deviation). Samples were similarly checked, and those with a genotyping success rate of <97% were removed. A principal component analysis performed with EIGENSTRAT version 3.0 had shown that German and Italian PEX cases and controls appeared to be well matched in terms of ancestry as well as in overall assessment (Supplementary Figs 12 and 13). Quality control criteria for Japanese and Italian data sets have been described

previously[64]. Both German and Italian genotypes were imputed into the 1,000 genomes data set (March 2012 haplotypes) using IMPUTE2 (ref. 68). Association analyses on the imputed data were performed with SNPTEST v2.5.2 (ref. 69).

A logistic regression model using gender and age as covariates in German and Italian samples and the first six principal components of genetic stratification in Japanese as previously described[22,64] was used to determine the per-allele, additive effect of the SNPs with a score-based, frequentist algorithm in SNPTEST2. By adding the dosages of rs3825942 and rs1048661 as additional covariates, we tested for independence of our 14 associated SNPs from the two SNPs (rs3825942 and rs1048661) of the original published locus. Regional association plots were created using LocusZoom version 1.1 (ref. 70). Statistical analysis for quality control, $\chi^2$ test of association, logistic regression for adjustment of $P$ values and multi-dimensional scaling for population stratification were performed using PLINK version 1.07 (ref. 71). The German and Italian haplotypes were estimated by PHASE version 2.1 (ref. 72). $P$ values and odds ratios (ORs) of different haplotypes were calculated with $\chi^2$-based tests using PLINK version 1.07.

**Human tissues.** Human donor eyes used for corneal transplantation with appropriate research consent were obtained from Caucasian donors and processed within 20 h after death. Informed consent to tissue donation was obtained from the donors or their relatives, and the protocol of the study was approved by the Ethics Committee of the Medical Faculty of the Friedrich-Alexander-Universität Erlangen-Nürnberg (No. 4218-CH) and adhered to the tenets of the Declaration of Helsinki for experiments involving human tissues and samples.

For RNA and DNA extractions, 52 donor eyes with manifest PEX syndrome without ($n = 43$) or with ($n = 9$) glaucoma (mean age, $81.1 \pm 7.5$ years; 32 female, 20 male) and 51 normal appearing age-matched control eyes (mean age, $76.3 \pm 10.2$ years; 23 female, 28 male) without any known ocular disease were used. For protein extractions, another 20 donor eyes without any known ocular disease (mean age, $74.3 \pm 5.54$ years; 11 female, 9 male) were used. The presence of characteristic PEX material deposits was assessed by macroscopic inspection of anterior segment structures and confirmed by electron microscopic analysis of small tissue sectors. Ocular tissues (Tenon's capsule, cornea, trabecular meshwork, iris, ciliary body, retina, choroid, lamina cribrosa and sclera) were prepared under a dissecting microscope and shock frozen in liquid nitrogen. Specimens of aortic wall were obtained at autopsy within 24 h after death from five organ donors with PEX syndrome (mean age, $71.6 \pm 3.5$ years) and five age-matched subjects without PEX (mean age, $68.3 \pm 7.7$ years).

**Genotyping of human tissues and cells.** DNA samples obtained from ocular tissues and cells were genotyped by Sanger sequencing. Primers surrounding the 14 intronic SNPs, rs8023330, rs1550436, rs2165241, rs28588430, rs28617339, rs4886778, rs8027022, rs2028386, rs4337252, rs12440667, rs12905253, rs11638944, rs12441130 and rs11631579, and the two exonic SNPs, rs1048661 and rs3825942 of the LOXL1 gene were designed using Primer3 software (http://frodo.wi.mit.edu/cgi-bin/primer3/primer3_www.cgi/) and supplied from Thermo Scientific (Schwerte, Germany) (Supplementary Table 3.1). Purified PCR fragments were sequenced using Big Dye Termination chemistry v.3.1 (Applied Biosystems, ABI, Weiterstadt, Germany) on an automated capillary sequencer (ABI 3730 Genetic Analyzer). All sequences were analysed using Sequencer5.1 (Gencodes) software and compared with the respective reference SNP sequences from the human genome sequence (hg19).

**Real-time PCR.** Ocular tissues and cultured cells were extracted using the Precellys 24 homogenizer and lysing kit together with the AllPrep DNA/RNA kit (Qiagen, Hilden, Germany) according to the manufacturer's instructions including an on column DNaseI digestion step using the RNase-free DNase Set (Qiagen). First-strand cDNA synthesis from 0.5 μg of total RNA was carried out using 200U Superscript II reverse transcriptase (Thermo Scientific) and 200 ng of random primers (Roche Life Science, Mannheim, Germany) in a 20 μl reaction. Quantitative real-time PCR was performed using the CFX Connect thermal cycler and software (Bio-Rad Laboratories, München, Germany). PCR reactions (25 μl) were run in duplicate and contained 2 μl of first-strand cDNA, 0.48 μM each of upstream- and downstream-primer, and SsoFast EvaGreen Supermix (Bio-Rad). Exon-spanning primers (Eurofins Genomics, Ebersberg, Germany), designed by means of Primer 3 software (available at: http://bioinfo.ut.ee/primer3/, accessed November 3, 2014), and PCR conditions are summarized in Supplementary Table 3.2. For quantification, serially diluted standard curves of plasmid-cloned cDNA were run in parallel, and amplification specificity was checked using melt curve and sequence analyses with the Prism 3100 DNA-sequencer (Applied Biosystems). For normalization of gene expression levels, mRNA ratios relative to the house-keeping gene GAPDH were calculated.

**Western blot analysis.** Total protein was extracted from iridal tissue specimens using the PRECELLYS24 tissue homogenizer (Bertin Instruments, Montigny-le-Bretonneux, France) and RIPA buffer (50 mM Tris-HCl pH 8.0, 150 mM NaCl, 1% NP-40, 0.5% DOC, 0.1% SDS) containing protease inhibitor cocktail (Sigma-Aldrich, Saint Louis, MO). Protein concentrations were determined by the Micro-BCA protein assay kit (Thermo Scientific). Proteins (10 μg per lane) were separated by 4–15% SDS–polyacrylamide gel electrophoresis under reducing conditions (1 × sample buffer: 75 mM Tris-HCl pH6.8, 2% SDS, 10% glycerol, 0.002% bromphenol blue, 100 mM DTT) and transferred onto nitrocellulose membranes (Bio-Rad) with the Trans-Blot Turbo transfer system (Bio-Rad) according to the manufacturer's recommendations. Membranes were blocked with SuperBlock T20 (Thermo Scientific) for 30 min and incubated for 1 h at room temperature with antibodies against LOXL1 (1 μg ml[−1]; kindly provided by Takako Sasaki, Erlangen), RXRα (2 μg ml[−1]; clone D-20, sc-553, Santa Cruz, Heidelberg, Germany) and ß-actin (0.5 μg ml[−1]; clone AC-15, A5441, Sigma-Aldrich) diluted in PBST-10% SuperBlock T20 (phosphate buffered saline pH 7.4, 0.1% Tween-20, 10% SuperBlock T20). In negative control experiments, the primary antibody was replaced by PBS. Immunodetection was performed with a horseradish peroxidase-conjugated goat anti-rabbit (31460, Thermo Scientific) or goat anti-mouse (405306, Biolegend, Fell, Germany) secondary antibody diluted in PBST-10% SuperBlock T20 at a final concentration of 5 ng ml[−1], and the Super Signal West Femto (LOXL1 and RXRα) or Pico (ß-actin) ECL kit (Thermo Scientific). Specific protein bands were quantitatively analysed with the LAS-3000 (Fujifilm, Düsseldorf, Germany) chemiluminescence detection system and software (Multi Gauge V1.1, Fujifilm). For normalization of LOXL1 protein expression levels, protein ratios relative to the house-keeping gene ß-actin were calculated.

**Plasmid construction.** The LOXL1 minimal promoter region was amplified as a 1,636 bp DNA fragment using the HotStar HiFidelity DNA polymerase (Qiagen) with primers and conditions as specified in Supplementary Table 3.3. Digestion with HindIII/NcoI resulted in a LOXL1 promoter fragment (73,925,346–73,926,784, NC_000015) extending from −1438 bp to the ATG start codon (+1) (ref. 35), which was cloned into the pGL4.10[luc2] reporter vector (Promega, Mannheim, Germany) via HindIII/NcoI resulting in a pGL4.10-LOXL1 construct.

To determine the influence of selected SNPs (1: rs8023330A > G, 2: rs1550436C > T, 3: rs2165241C > T, 4: rs28588430G > C, 5: rs28617339C > T, 6: rs4886778A > C, 7: rs8027022A > G, 8: rs2028386G > C, 9: rs4337252C > G, 10: rs12440667C > T, 11: rs12905253G > A, 12: rs11638944C > G, 13: rs12441130T > C and 14: rs11631579A > G) on LOXL1 promoter activity, synthetic DNA fragments (Eurofins Genomics) harbouring the risk alleles (14sR) or the non-risk alleles (14sN) of all 14 SNPs [EcoRV-Bgl II-SNPs1 to 14-Bgl II-HindIII] were cloned into pGL4.10-LOXL1 via EcoRV/HindIII upstream of the LOXL1 promoter resulting in pGL4.10-LOXL1-14sR/N constructs (Fig. 5a). The DNA flanking sequences surrounding each of the 14 SNPs consisted of 51 bp of genomic DNA sequence (NC_000015) with each SNP in the center. The basal LOXL1 promoter construct was prepared by restriction of pGL4.10-LOXL1-14sR with Bgl II. Synthetic DNA fragments containing risk and non-risk alleles, respectively, [EcoRV-EcoRI-SNPs 1 to 5-EcoRI-NdeI-SNPs 6 to 10-NdeI-Bgl II-SNPs 11 to 14-Bgl II-HindIII] were subsequently cloned into the EcoRV/HindIII sites of pGL4.10-LOXL1, and digested with EcoRI, NdeI, and Bgl II, respectively, to obtain deletion constructs pGL4.10-LOXL1-14sR/N1-5, -14sR/N6-10 and -14sR/N11-14. Genomic sequences (228–292 bp) including single SNPs with 5′-KpnI and 3′-Bgl II (SNPs 11, 12 and 14) or 5′-KpnI and 3′-BamHI (SNP 13) restriction sites, respectively, were amplified by PCR with primers specified in Supplementary Table 3.3 and cloned into the basal LOXL1 promoter construct via KpnI/Bgl II resulting in pGL4.10-LOXL1-11R/N, -12R/N, -13R/N and -14RN constructs.

Digestions were carried out with enzymes from NEB (Frankfurt, Germany), ligations with the Rapid DNA Dephos & Ligation kit (Roche), and plasmid preparations with the Nucleobond Xtra Maxi EF kit (Macherey-Nagel, Düren, Germany). Confirmatory sequencing was carried out for all constructs (Eurofins Genomics). All sequences of pGL4.10 constructs and the plasmids themselves are available upon request.

**Cell culture and dual luciferase reporter assays.** Tenon's capsule biopsies were obtained from five German patients with PEX syndrome (mean age, $78.3 \pm 10.5$ years) and from 25 age-matched patients without PEX (mean age, $73.0 \pm 5.4$ years) during cataract surgery. In addition, ten biopsy samples were obtained from younger German patients (mean age, $18.2 \pm 6.4$ years) during vitreoretinal surgery, and from six Japanese patients without PEX (mean age, $66.3 \pm 4.5$ years) during pterygium surgery. Informed consent to tissue donation was obtained from the patients, and the protocol of the study was approved by the Ethics Committee of the Medical Faculty of the Friedrich-Alexander-Universität Erlangen-Nürnberg (No. 4218-CH) and the Institutional Review Board Committee of Ozaki Eye Hospital (No. 16). The experiments adhered to the tenets of the Declaration of Helsinki for experiments involving human tissues and samples.

Primary hTCF cultures were established from Tenon's capsule biopsies and maintained in Dulbeccós modified Eaglés medium (DMEM/Haḿs F12; Invitrogen, Darmstadt, Germany) containing 15% (v/v) fetal bovine serum (FBS) and antibiotic-antimycotic solution (PSA; Invitrogen) at 37 °C in a humidified 95% air–5% $CO_2$ atmosphere. Primary hONHA cultures were generated from lamina cribrosa tissue of five normal donor eyes (mean age, $63.7 \pm 9.5$ years) and maintained in DMEM/Ham's F12 with 10% FBS, 1% PSA, supplemented with 5 ng ml[−1] bFGF and 5 ng ml[−1] PDGF-AA (Sigma-Aldrich), in a 95% air–5% $CO_2$ humidified atmosphere at 37 °C. Primary hLEPC cultures were generated from

corneoscleral donor tissue ($n = 5$; mean age, $64.7 \pm 7.3$ years) with appropriate research content provided by the Erlangen Cornea Bank after corneal endothelial transplantation and maintained in Keratinocyte serum free medium (Invitrogen) supplemented with bovine pituitary extract, epidermal growth factor and PSA. Primary hTMC from three different donors were obtained from Provitro (Berlin, Germany) and grown in DMEM supplemented with 10% FBS and 1% PSA. Primary hASMC were also obtained from Provitro and grown in SMCM medium (Provitro). For transfection experiments, primary cells were used at passages 3–4. The immortalized hNPEC line ODM-2 was kindly supplied by Miguel Coca-Prados (Fundación de Investigación Oftalmológica, Oviedo, Spain), grown in DMEM (with $4.5 \, \mathrm{g \, l^{-1}}$ glucose) supplemented with 10% FBS and $50 \, \mu\mathrm{g \, ml^{-1}}$ gentamycin, and was used up to passage 12. HEK293T cells and the mouse embryonic fibroblast 3T3 cell line were obtained from ATCC (Manassas, VA) and grown in DMEM (with $4.5 \, \mathrm{g \, l^{-1}}$ glucose) supplemented with 10% FBS and 1% PSA.

Cells ($1.0 \times 10^6$) were transiently transfected by electroporation using the Nucleofector II transfection device (Lonza, Köln, Germany) and the Amaxa Basic Fibroblasts Nucleofector kit (Lonza) for hTCF, hTMC, hONHA, hASMC and 3T3 cells or the Amaxa Cell Line V Nucleofector kit for hNPEC, and hLEPC, and $6 \, \mu\mathrm{g}$ pGL4.10-LOXL1 reporter construct together with $0.9 \, \mu\mathrm{g}$ Renilla luciferase plasmid pGL4.74 (Promega). Nucleofector programs used were U-023 for hTCF, hTMC, hONHA, hNPEC and 3T3 cells, U-025 for hASMC and V-023 for hLEPC. Transfected cells were seeded into six-well plates in triplicate. HEK 293 T cells were seeded in six-well plates at a density of $2.5 \times 10^5$ cells per well and transfected in triplicate with $0.5 \, \mu\mathrm{g}$ pGL4.10-LOXL1 reporter construct together with $0.1 \, \mu\mathrm{g}$ pGL4.74 using Lipofectamine LTX (Thermo Scientific) according to the manufacturer's recommendations. At 24 h post-transfection, cells were washed with PBS and lysed using Glo Lysis Buffer (Promega).

LOXL1 promoter activity of cell extracts was assayed using the Dual-Glo Luciferase Assay System (Promega), and luminescence was measured on a GloMax luminometer (Promega). To correct for transfection efficiency, luciferase reporter construct activity was normalized to Renilla luciferase activity, and results are expressed as the ratio of Firefly luciferase [luc2] to Renilla luciferase. Data represent at least five independent experiments for each cell type analysed.

In different sets of experiments, cells were co-transfected with RXRα-specific siRNA (see below) or stimulated with all trans-retinoic acid (Sigma-Aldrich) at a concentration of $2 \, \mu\mathrm{M}$ for 48 h.

**Allele-specific expression analysis.** hTCF cell lines ($n = 15$) heterozygous for the risk and non-risk alleles of LD-SNPs 11–14 were analysed using a predesigned TaqMan SNP Genotyping Assay for SNP 13 (rs12441130T>C) (C_30687884_10, Applied Biosystems) to compare the relative allelic expression of LOXL1 pre-mRNA. Real time PCR based SNP detection was performed using $2 \, \mu\mathrm{l}$ of cDNA preparation transcribed with random primers in a QuantStudio 12 K Flex Real-Time PCR System machine (Applied Biosystems) according to the manufacturer's instructions. The detection difference of the two alleles was normalized using genomic DNA from each donor. The relative expression of the disease-associated allele compared with the other allele was analysed in triplicates and determined using the $\Delta\Delta$Ct method. Genomic DNA and pre-mRNA from cell lines known to be homozygous for the risk ($n = 4$) or non-risk alleles ($n = 4$) of these SNPs was included as a reference.

**Chromatin immunoprecipitation.** ChIP experiments were performed on hTCF cell lines ($n = 2$) heterozygous for the risk and non-risk alleles of LD-SNPs rs12905253, rs11638944, rs12441130 and rs11631579 using antibodies against RNA polymerase II (Santa Cruz, sc-899) or antibodies against RXRα (Santa Cruz, sc-553: clone D-20 and sc-46659: clone F-1). Antibodies against histone H3 (Abcam, Cambridge, UK, ab1791) and acetylated histone H3K27Ac (Diagenode, Denville, NJ, pAb-174-050) were used as positive controls and normal rabbit IgG (Merck Millipore, Darmstadt, Germany, 12-370) as negative control. Briefly, cells were grown in 15 cm dishes to about 80% confluency before being subjected to 1% formaldehyde for 12 min at 4 °C to crosslink protein complexes to DNA. Cross linking was quenched by addition of 125 mM glycine. Cells were lysed using ChIP lysis buffer (1% SDS, 10 mM EDTA, 50 mM Tris, $40 \, \mu\mathrm{l \, ml^{-1}}$ cOmplete protease inhibitor cocktail) (Sigma-Aldrich). Cell lysates were sonicated in 15 s pulses for a total of 14 min (Diagenode, Bioruptor Plus sonication device), diluted with ChIP dilution buffer (0.01% SDS, 1.1% Triton X-100, 1.2 mM EDTA, 16.7 mM Tris, 167 mM NaCl), and precipitated with $2 \, \mu\mathrm{g \, ml^{-1}}$ antibodies and 25% protein A agarose (Merck Millipore). After incubation and centrifugation, the pellet was resolved in low-salt buffer (0.1% SDS, 1% Triton X-100, 2 mM EDTA, 20 mM Tris, 150 mM NaCl) and washed in high-salt buffer (0.1% SDS, 1% Triton X-100, 2 mM EDTA, 20 mM Tris, 500 mM NaCl), LiCl buffer (0.25 M LiCl, 1% Igepal, 1% deoxycholate, 1 mM EDTA, 10 mM Tris) and TE buffer (1 mM EDTA and 10 mM Tris). ChIP elution buffer (0.0841 g NaHCO₃, 1% SDS) was used to resolve chromatin antibody binding. To break up cross-links between proteins and DNA, probes, samples were sequentially incubated in 5 M NaCl and in 0.5 M EDTA, 1 M Tris and $40 \, \mu\mathrm{g}$ Proteinase K. After phenol–chloroform extraction, the precipitated DNA was dissolved in Nuclease-free $H_2O$. Samples were analysed by qPCR using the TaqMan SNP Genotyping Assay for SNP 13.

**Splicing analysis.** Alternative mRNA splicing was analysed in hTCF cell lines homozygous for the risk ($n = 8$) or non-risk ($n = 8$) alleles of LD-SNPs 11–14 by qPCR using primers designed for exon 1 and the additional exon 1A included in the alternative LOXL1 transcript LOXL1-002 (LOXL1_E1A) (ENST00000566011; www.ensembl.org) known to undergo NMD (Fig. 7a, Supplementary Table 3.3). For inhibition of NMD, cells were treated with $200 \, \mathrm{ng \, ml^{-1}}$ puromycin (Sigma-Aldrich) for 10 h.

**Electrophoresis mobility (super)shift assays.** A nuclear extraction kit (NE-PER Nuclear and Cytoplasmic Extraction Reagents, Thermo Scientific) was used to prepare cytoplasmic and nuclear extracts from primary hTCF, hTMC, hNPEC, hLEPC, hASMC, HEK293T and 3T3 cells according to the manufacturer's instructions.

Biotinylated DNA probes were amplified by PCR using 5′-biotinylated 20–23 bp primers (Supplementary Table 3.3) (Eurofins Genomics) using tissue samples from genotyped individuals homozygous for the risk and non-risk haplotypes as templates. All binding reactions were carried out using the non-radioactive LightShift Chemiluminescent EMSA Kit (Thermo Scientific) according to manufacturer's protocol. Briefly, the binding reaction mixture (total volume $20 \, \mu\mathrm{l}$) containing $2 \, \mu\mathrm{l}$ of nuclear extract (at a concentration of $\sim 2 \, \mu\mathrm{g \, \mu l^{-1}}$) was pre-incubated with $50 \, \mathrm{ng \, \mu l^{-1}}$ of poly [d(I-C)] in binding buffer (10 mM Tris, 50 mM KCl, 1 mM DTT, pH 7.0) on ice for 10 min. After addition of 20 fmol biotin-labeled target DNA, the binding reaction was incubated at room temperature for 20 min. For competition experiments, a 200-fold molar excess (4 pmol) of unlabelled oligonucleotides were included in the pre-incubation mixture. Subsequently, all reactions were loaded onto a 4% (supershift) or 6% (EMSA) non-denaturing polyacrylamide gel and run for 1 h in $0.5 \times$ TBE running buffer (45 mM Tris, 45 mM boric acid, 1 mM EDTA, pH 8.3). After electrophoresis, complexes were transferred to a positively charged nylon membrane, ultraviolet crosslinked at $120 \, \mathrm{mJ \, cm^{-2}}$, probed with streptavidin-HRP conjugate and incubated with enhanced chemiluminescence substrate (ECL; Thermo Scientific). Gel shifts were quantitatively analysed with the LAS-3000 (Fujifilm, Düsseldorf, Germany) chemiluminescence detection system and software (Multi Gauge V1.1, Fujifilm).

For supershift assays, $1 \, \mu\mathrm{g}$ of antibodies against NFκB p50 (sc-7178X), NFκB p65 (sc-7151X), PPARα (sc-9000X), PPARγ (sc-7196X), NR2F1 (sc-6575X), HNF-4α (sc-8987X), myogenin (sc-13137X), ROAZ (sc-514748X), NF1 (sc-74445), RXRα (sc-46659X), RXRβ (sc-742X), RXRγ (sc-365252 X), RARα (sc-551X), LXRα-β (sc-13068X), VDR ( sc-9164), TRα-β (sc-772X) and ERα (sc-8005X), all purchased from Santa Cruz, and against ZID/ZBTB6 (ab67628) purchased from Abcam was added to the reaction mixture and incubated at room temperature for 30 min prior to addition of the biotinylated probe.

Recombinant proteins of full-length human RXRα (TP310311, OriGene, Rockville, MD), DNA-binding domain (aa 11-228; pro-437, Prospec, Rehovot, Israel) and ligand-binding domain (aa 198–462; 31135, Active Motif, Carlsbad, CA) of RXRα as well as a RXR consensus oligonucleotide (sc-2547, Santa Cruz) were used as positive controls.

**siRNA silencing.** hTCF ($0.75 \times 10^6$) were transiently transfected with specific siRNA (ON-TARGETplus SMARTpool, GE Healthcare Dharmacon, Freiburg, Germany) for UPF1 (75 pmol; target sequences: CAGCGGAUCGUGUGAAGAA, CAAGGUCCCUGAUAAUUAU, GCAGCCACAUUGUAAAUCA, GCUCGC AGACUCUCACUUU) or RXRα (150 pmol; target sequences: GCGCCAUCGU CCUCUUUAA, GCAAGGACCGGAACGAGAA, AGACCUACGUGGAGGC AAA, UCAAAUGCCUGGAACAUCU) by electroporation using the Nucleofector II transfection device (Lonza) and the Amaxa Basic Fibroblasts Nucleofector Kit (Lonza) with the nucleofector programme U-023. Transfections with scrambled siRNA (ON-TARGETplus Non-targeting pool, GE Healthcare Dharmacon) served as controls. Transfected cells were seeded into six-well plates in duplicate and collected at 48 h post transfection for real-time PCR analysis.

**In silico analysis.** The rSNP-MAPPER software tool (available at http://geno-me.ufl.edu/mapper) was used to predict transcription factor binding sites affected by the analysed SNPs[37]. The sequences of the vectors containing the risk or non-risk alleles, respectively, were provided as input. Furthermore, predictions were also obtained for the genomic regions of each single SNP. UCSC Genome Browser (http://genome.ucsc.edu/) was used to visualize the analysed SNPs in the genomic context and to explore results of ENCODE experiments to identify regions with evidence of potential regulatory activity.

**Statistical analysis.** Group comparisons for the reporter assays, expression analyses, and comparative allele representation were performed using an unpaired two-tailed $t$-test or a Mann–Whitney $U$-test using SPSS v.20 software (IBM, Ehningen, Germany). $P < 0.05$ was considered statistically significant.

**Data availability.** The genome-wide association SNP results are available upon request by contacting Francesca Pasutto at Francesca.pasutto@uk-erlangen.de. Any additional data (beyond those included in the main text and Supplementary

Information) that support the findings of this study are also available from the corresponding author upon request.

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

## Acknowledgements

This work was supported by the German Research Foundation (SCHL 366/8-1 and SFB-539) and the Interdisciplinary Center for Clinical Research (IZKF) of the University Hospital, Erlangen, Germany. We would like to thank Dr Miguel Coca-Prados, Fundación de Investigación Oftalmológica, Oviedo, Spain, for providing the human nonpigmented ciliary epithelial cell line ODM-2, Ekaterina Gedova, Myriam Eitl, Jasmine Onderka, and Victoria Lauer for excellent technical support, as well as Prof. Dr Julio Vera and Martin Eberhardt from the Department of Dermatology for discussions on in silico prediction tools for transcription factor binding sites.

## Author contributions

F.P., M.Z., U.H., D.B., S.U., F.F., P.L. and U.S.-S. designed, performed, analysed and interpreted experiments and wrote the manuscript. J.S., T.A. and C.C.K. designed, performed, analysed and interpreted experiments. M.O., D.P., P.F., T.M., S.N., T.K., S.M., E.S., P.M., D.C., C.G. and H.-E.W. contributed patients' samples and clinical data. F.E.K. and A.R. interpreted results and edited the manuscript.

## Additional information

**Competing interests:** The authors declare no competing financial interests.

**Reprints and permission** information is available online at http://npg.nature.com/ reprintsandpermissions/

