## [Peer Review File · Nature Communications]

Reviewers' Comments:

Reviewer #1 (Remarks to the Author)

Pasutto and colleagues present a series of thorough and elegant studies demonstrating a functional mechanism through which common genetic variants contribute to disease. The study is focused on the LOXL1 locus and its relationship to pseudoexfoliation syndrome (PEX). Not only does PEX lead to severe glaucoma with poor outcomes, it increases the risk of complications from cataract surgery and has clear links with connective tissue dysfunction in non-ocular tissues.

This study aimed to determine a functional mechanism through which common SNPs at the LOXL1 locus contribute to disease. The authors begin with a fine-mapping study which defines the region of interest and selects 14 SNPs genotyped in 3 populations by imputation, which define a "risk" and a "normal" haplotype. A minor limitation is that these haplotypes were not re-sequenced in affected individuals to identify all possible variants on these haplotypes in patients, but relied on imputation with reference to the 1000 genomes project and therefore only common variants represented in 1000 genomes in strong LD with the tag SNPs were selected.

The authors perform a series of functional experiments, showing that the risk haplotype decreased LOXL1 transcript levels in multiple relevant ocular tissues and cell lines and highlight a series of 4 SNPs that appear responsible for this. Analysis of the 4 individual SNPs shows that at least 2 of them, when considered in isolation, actually increase transcription, consistent with the enhancer marks observed in the region. This seemingly contradictory result is explained by experiments showing that the risk alleles increase expression of an alternatively spliced transcript, which is then subject to nonsense mediated decay, leading to an overall decrease in transcript level. Further, they show that RXRalpha is the transcription factor responsible for activating transcription of the alternative transcript.

This is the first report demonstrating a plausible functional mechanism for non-coding variants at this locus and although there are limitations as outlined by the authors themselves, it is an outstanding study that explains for the first time, the extremely strong association signals observed at this locus. It is unable to untangle the flipped allele phenomenon observed between Caucasian and Asian populations and this will require further context specific research. Targeted resequencing of the risk haplotypes in both populations may assist with this.

This is a very well written and convincing paper of interest to both ophthalmology and functional genetics in general. I have only a few minor suggestions:

1. In Table S2, please indicate which are the coding SNPs in the haplotype and where they sit in relation to the 14 non-coding SNPs listed in Table S1.
2. Please list the ocular tissues used in the methods section
3. It is not immediately clear why the optic nerve head astrocytes were chosen. Presumably they represent the lamina cribosa?
4. Page 13 line 322. Change "This data..." to "These data..."
5. Some figure titles use "German" (eg Figure 2) and some use "DE" (eg Figure 1). Please be consistent. Also, the term "unconditional" in the title of Figure 1 is not easily interpreted without going through all the methods/figures to find that a conditional association analysis was also performed.

Reviewer #2 (Remarks to the Author)

The goal of the study as stated by the authors was to find an explanation for how noncoding regulatory variants in LOXL1 affect LOXL1 transcription and expression thereby conferring susceptibility to PEX as a common complex disease. Since there is allele flipping in the Japanese

that occurs in the intronic region the authors explore, they have not achieved the objective of determining how the allelic changes contribute to XFS and actually indicate as much in the limitation section of their discussion. In the discussion they state, "Although this enigma cannot be solved by the present study, potential regulatory differences at the DNA level as well as potential differences in LOXL1 expression quantitative phenotypes and their relation to variations in PEX prevalence between European and Japanese populations will be the subject of future studies." Thus, while the authors have done a good deal of work to explain how LOXL1 expression is regulated via polymorphic changes in intron 1 and intron 2, they have not closed the gap regarding how LOXL1 gene variants contribute to XFS. They need to revise their last sentence of the abstract to indicate that while new mechanisms for regulating LOXL1 gene expression were uncovered their relation to XFS remains unknown.

On the basis of a genome-wide association study in a German cohort of PEX patients, the authors selected a cluster of 14 common SNPs within introns 1 and 2 of LOXL1 in complete linkage disequilibrium (LD) with the known variants, and confirmed their association with PEX in European and Asian populations. Importantly, all 14 SNPs were significantly associated with PEX syndrome / glaucoma in a Japanese cohort (1484 cases and 1188 controls), although risk alleles were completely reversed between the two European and Japanese populations. However, in all three cohorts analyzed, the 14 selected variants are located within the most frequent haplotype representing the risk haplotype in each specific population (haplotype 4). While considerable functional studies follow, they cannot overcome the issue that the functional changes do not impact disease when the opposite allele is associated with XFS in Asians.

Using models of disease-relevant cell types, the authors go on to provide experimental evidence for a functional variant, rs11638944:C>G, located in a genomic region with regulatory potential downstream of the canonical LOXL1 promoter, to exert allele-specific effects on LOXL1 transcriptional activity through differential transcription factor binding and alternative pre-mRNA splicing in a cell type-specific manner. They show quite elegantly that increased transcriptional activity at the risk sequence is associated with reduced binding of retinoid X receptor alpha (RXR α) and with enhanced unproductive splicing coupled to nonsense-mediated decay (NMD), which reduces the final steady state levels of LOXL1 mRNA in cells and tissues of risk allele carriers. But again they cannot lay claim that any of these findings "predisposes to connective tissue alterations and clinical complications" and this statement must be removed from the manuscript due to the allele flipping issue.

Specific issues:

1. Abstract, line 9: The phrase "Among those four common variants", seems to be out of context and should be clarified.
2. Line 90 "p=7.62-22" – this is a mistake and should be corrected.
3. Supplemental Table1 - Some of the column headers are cutoff and are not labeled. This Table1 should be revised.
4. The conditional analysis related to the 14 intronic LOXL1 SNPs is troubling and somewhat confusing. When the authors control for the known major LOXL1 variants previously published (rs3825942 and rs11852604), most of these 14 SNPs are not significant. Thus the conditional analysis indicates there is nothing special about any of these 14 SNPs based on a priori evidence. Furthermore in the Italian cohort rs11852604 is significant in the conditional analysis, yet the potential significance of this SNP is virtually ignored. The authors need to explain why they focused on these 14 SNPs – did they basically assume this intronic region must have regulatory elements? In this instance the GWAS data does not add to what we already know. A lack of controlling for age in the Italian cohort is another concern with the GWAS data.
5. The authors assessed whether the 14 SNP risk haplotype correlated with tissue expression

levels of LOXL1. DNA samples of various post-mortem ocular tissues isolated from 52 individuals with manifest PEX syndrome/glaucoma and 51 individuals without PEX were used for genotyping the 14 intronic SNPs, while mRNA expression levels of LOXL1 were analyzed by real time qRT-PCR.

The source and ethnic makeup of this material is not provided. I suspect the material came from Caucasians but it would be interesting if material from Asian patients were studied.

6. Line 240: Human TCF treated with puromycin, which is known to selectively stabilize NMD transcripts increased the level of LOXL1-E1A 5-fold ($p < 0.001$), whereas levels of wild-type LOXL1 mRNA were decreased 4-fold ($p < 0.0001$). What was the haplotype of these cells?

7. The labels for the DNA probes S11R and S11N in supplemental figure 5 are slightly cutoff and should be revised.

8. Line 316: A new paragraph should not start with the phrase, "In fact".

9. The authors report, RXRa was found to be expressed in nuclear fractions of all disease-relevant cell types including hTCF, hTMC, hNPEC and hASMC. Were non-relevant cells tested and if so was the result any different?

10. Line 352: the word "fragmentary" should probably be replaced with "rudimentary".

11. Please use 1350 or 1,350 but not 1.350 to describe the number of patients. I have a similar comment for line 483 and 484.

12. The abbreviation EMSA in the legend for figure 7 should be spelled out.

Point-by-point response:

Reviewer #1:

1. *A minor limitation is that these haplotypes (“risk” and “normal”) were not re-sequenced in affected individuals to identify all possible variants on these haplotypes in patients, but relied on imputation with reference to the 1000 genomes project and therefore only common variants represented in 1000 genomes in strong LD with the tag SNPs were selected.*

.....

It is unable to untangle the flipped allele phenomenon observed between Caucasian and Asian populations and this will require further context specific research. Targeted resequencing of the risk haplotypes in both populations may assist with this.

Response: We entirely concur with the reviewer concerning the need for further large scale sequencing studies to identify causal variants across populations, which we expect to be rare and not detectable through this GWAS approach. Such analyses are of sufficient complexity and dimension to be performed in a separate study, particularly in view of the considerable volume of the present study. On the basis of the present findings, we are indeed planning to perform targeted resequencing of the risk haplotype in Caucasian and Asian populations in a separate study, and we have mentioned the need for such a study in the last paragraph of the Discussion (lines 465-468).

3. *In Table S2, please indicate which are the coding SNPs in the haplotype and where they sit in relation to the 14 non-coding SNPs listed in Table S1.*

Table S2 has been modified accordingly and indicates the coding SNPs in bold and underlined types. We have also included the coding SNPs and their position in Table S1, as suggested by Reviewer 2.

4. *Please list the ocular tissues used in the methods section.*

This information has been included (page 21, line 528).

5. *It is not immediately clear why the optic nerve head astrocytes were chosen. Presumably they represent the lamina cribrosa?*

Response: Primary human optic nerve head astrocyte cultures were in fact generated from lamina cribrosa tissue, as stated in the methods section (page 24, line 615). Because we understand the lamina cribrosa as part of the optic nerve head, we just applied the more general and more common term “optic nerve head astrocytes”.

6. *Page 13 line 322. Change “This data...” to “These data...”*

This has been changed (line 323).

7. *Some figure titles use “German” (eg Figure 2) and some use “DE” (eg Figure 1). Please be consistent. Also, the term “unconditional” in the title of Figure 1 is not easily interpreted without going through all the methods/figures to find that a conditional association analysis was also performed.*

Figure titles have been modified accordingly, now only using “German” or “Italian” terms. According to the reviewer’s suggestion, we also removed the term “unconditional” from Figures 1 and S1, because the conditional analysis is mentioned in the text only afterwards (line 115).

Reviewer #2:

1. *The goal of the study as stated by the authors was to find an explanation for how noncoding regulatory variants in LOXL1 affect LOXL1 transcription and expression thereby conferring susceptibility to PEX as a common complex disease. Since there is allele flipping in the Japanese that occurs in the intronic region the authors explore, they have not achieved the objective of determining how the allelic changes contribute to XFS and actually indicate as much in the limitation section of their discussion. In the discussion they state, “Although this enigma cannot be solved by the present study, potential regulatory differences at the DNA level as well as potential differences in LOXL1 expression quantitative phenotypes and their relation to variations in PEX prevalence between European and Japanese populations will be the subject of future studies.” Thus, while the authors have done a good deal of work to explain how LOXL1 expression is regulated via polymorphic changes in intron 1 and intron 2, they have not closed the gap regarding how LOXL1 gene variants contribute to XFS. They need to revise their last sentence of the abstract to indicate that while new mechanisms for regulating LOXL1 gene expression were uncovered their relation to XFS remains unknown.*

Response: We agree with the reviewer and apologize for having overstated our findings. The last sentence of the abstract has been revised to reflect that our haplotypic and functional analyses only uncovered new functional mechanisms through which common noncoding variants influence *LOXL1* expression.

2. On the basis of a genome-wide association study in a German cohort of PEX patients, the authors selected a cluster of 14 common SNPs within introns 1 and 2 of LOXL1 in complete linkage disequilibrium (LD) with the known variants, and confirmed their association with PEX in European and Asian populations. Importantly, all 14 SNPs were significantly associated with PEX syndrome / glaucoma in a Japanese cohort (1484 cases and 1188 controls), although risk alleles were completely reversed between the two European and Japanese populations. However, in all three cohorts analyzed, the 14 selected variants are located within the most frequent haplotype representing the risk haplotype in each specific population (haplotype 4). While considerable functional studies follow, they cannot overcome the issue that the functional changes do not impact disease when the opposite allele is associated with XFS in Asians.

Response: We concur with the reviewer's comments that the considerable functional biological experiments we undertook are unable to overcome the issue that these changes are unlikely to impact on disease when the alleles are flipped. See also our response above and below. We wish to emphasize a few points here, nonetheless:

- The relationship between *LOXL1* common variants and PEX is, according to our knowledge, the only example known to date of a true, reproducible allele reversal phenomenon. Although the phenomenon of allele flipping/reversal has been discussed and hypothesized (<https://www.ncbi.nlm.nih.gov/pmc/articles/PMC1821115/> Lin PI et al., AJHG 2007), the closest example to an allele flip (described by Sleiman PM et al., N Engl J Med 2009) observed in asthma genetics was ultimately proven to be a false positive (Torgerson DG et al., Nat Genet 2011).
- Our study robustly establishes this clearly not just with single SNP analysis, but also with entire longer range haplotypes.
- We now go further than all other groups studying the genetics of *LOXL1* to show a clear effect exerted by these strongly associated common variants on *LOXL1* expression.
- This would hopefully stimulate further work on linking the *LOXL1* common variants, *LOXL1* gene expression, and susceptibility to PEX as well as targeted resequencing of the risk haplotype in Caucasian and Asian populations upon publication of the data. We also point to the need for such studies on page 19, lines 465-468.

3. Using models of disease-relevant cell types, the authors go on to provide experimental evidence for a functional variant, rs11638944:C>G, located in a genomic region with regulatory potential downstream of the canonical LOXL1 promoter, to exert allele-specific effects on LOXL1 transcriptional activity through differential transcription factor binding and alternative pre-mRNA splicing in a cell type-specific manner. They show quite elegantly that increased transcriptional activity at the risk sequence is associated with reduced binding of retinoid X receptor alpha (RXR α) and with en-

hanced unproductive splicing coupled to nonsense-mediated decay (NMD), which reduces the final steady state levels of LOXL1 mRNA in cells and tissues of risk allele carriers. But again they cannot lay claim that any of these findings “predisposes to connective tissue alterations and clinical complications” and this statement must be removed from the manuscript due to the allele flipping issue.

Response: We agree with the reviewer and again apologize for having overstated our findings. Such misinterpretations have been removed from the last sentence of the abstract, and similar phrases in the text (lines 68, 80, 457, 465-468) have been modified accordingly (all highlighted in yellow).

4. Abstract, line 9: The phrase “Among those four common variants”, seems to be out of context and should be clarified.

Response: The phrase “four common variants” refers to the “3.5-kb four-component polymorphic locus positioned in intron 1 and 2 of LOXL1” in line 8.

5. Line 90 “p=7.62-22” – this is a mistake and should be corrected.

We have corrected this to “p=7.62E-22” (line 90).

6. Supplemental Table 1 - Some of the column headers are cutoff and are not labeled. This Table 1 should be revised.

Table 1 has been revised accordingly.

7. The conditional analysis related to the 14 intronic LOXL1 SNPs is troubling and somewhat confusing. When the authors control for the known major LOXL1 variants previously published (rs3825942 and rs11852604), most of these 14 SNPs are not significant. Thus the conditional analysis indicates there is nothing special about any of these 14 SNPs based on a priori evidence. Furthermore in the Italian cohort rs11852604 is significant in the conditional analysis, yet the potential significance of this SNP is virtually ignored. The authors need to explain why they focused on these 14 SNPs – did they basically assume this intronic region must have regulatory elements? In this instance the GWAS data does not add to what we already know. A lack of controlling for age in the Italian cohort is another concern with the GWAS data.

Response: As the reviewer correctly notes, there is little evidence for a new independent effect captured by any of the 14 SNPs analyzed, but this was actually in line with our expectations, because all 14 intronic SNPs were in complete LD with the coding SNP rs3825942. Hence, adjusting for the allele dosage of rs3825942 abolishes any evidence of *independent* association at the remaining 14 SNPs. However, the aim of the conditional analysis was not to show independence from previously identified SNP associations, but just to test for this unlikely possibility and to take a further step forward towards unraveling the still unknown functional background of the previously reported associations. Therefore, we explored the possibility that other SNPs in LD with the known variants could have functional biological consequences, and selected these 14 SNPs on the basis

of their significance of association (in LD with rs3825942) and their location within putative regulatory regions (as mentioned in lines 101 and 366).

SNP rs11852604: Even though rs11852604 appears to show independent association in Italian patients, a closer look at its minor allele frequency (MAF<0.3%, no minor allele homozygotes at all) indicates that this very low frequency might explain the high p-value of rs11852604. In addition, despite an elevated concordance, we observed a lower coefficient of correlation ($r^2 = 0.172$) for this SNP. Usually, such SNPs are filtered out in genome wide analyses. Thus, the great rarity of this variant and the lack of replication in the larger German population already argued against a potential significance of this SNP within the scope of the present study. Moreover, in line with the reviewer's concerns, we performed refined association and conditional analyses for the Italian cohort (updated Supplementary Figures S1 and S2B) after having collected the missing information on age and gender. In addition, we performed a principal component analysis (PCA) for the Italian cohort showing a well-matched data set after exclusion of 21 samples with PC1 <0.02 (new Supplementary Figure S9). The updated Supplementary Figure S1 now shows the regional association plot for the Italian dataset using a logistic regression model adjusted for age and gender, as has been already shown for the German dataset. As expected, this repeated analysis confirmed the initial associations but resulted in lower p-values, which corresponded to those of the larger German dataset (updated Table 1). In the repeated conditional analysis, SNP rs11852604 was no longer significant, which may be explained by removal of samples with PC1 <0.02 as described above.

8. The authors assessed whether the 14 SNP risk haplotype correlated with tissue expression levels of LOXL1. DNA samples of various post-mortem ocular tissues isolated from 52 individuals with manifest PEX syndrome/glaucoma and 51 individuals without PEX were used for genotyping the 14 intronic SNPs, while mRNA expression levels of LOXL1 were analyzed by real time qRT-PCR. The source and ethnic makeup of this material is not provided. I suspect the material came from Caucasians but it would be interesting if material from Asian patients were studied.

Response: All tissue samples were in deed derived from Caucasian donors (added on page 21, line 520). The only material studied, that was derived from Japanese patients, were Tenon's capsule biopsies used for isolation and cultivation of fibroblasts (line 612) and subsequent luciferase assays (line 178 and Figure S4A), in order to test any modulating effects of ethnicity on transcriptional activity of LOXL1. Further studies are planned to perform genotype-phenotype correlations in a larger collection of tissue samples from Asian patients.

9. Line 240: Human TCF treated with puromycin, which is known to selectively stabilize NMD transcripts increased the level of LOXL1-E1A 5-fold ($p<0.001$), whereas levels of wild-type LOXL1 mRNA were decreased 4-fold ($p<0.0001$). What was the haplotype of these cells?

Response: Alternative mRNA splicing was analyzed in hTCF cell lines homozygous for the risk (n=8) or non-risk (n=8) alleles of LD-SNPs 1-14 (corresponding to the risk haplotype 4 and the non-risk haplotypes 1 and 5 in Supplemental Table S2). Whereas the general effect of puromycin

treatment on LOXL1 transcript levels (Figure 6D) was analyzed in all 16 cell lines, genotype-correlated basal expression levels of LOXL1 transcripts were assessed in risk- and non-risk haplotypes separately, without puromycin treatment of cells (Figure 6E). This information has been added to the Figure legend.

10. The labels for the DNA probes S11R and S11N in supplemental figure 5 are slightly cutoff and should be revised.

This has been performed.

11. Line 316: A new paragraph should not start with the phrase, "In fact".

"In fact" has been deleted.

12. The authors report, RXR α was found to be expressed in nuclear fractions of all disease-relevant cell types including hTCF, hTMC, hNPEC and hASMC. Were non-relevant cells tested and if so was the result any different?

Response: We have actually tested HEK293 cells in addition, which showed similar expression of RXR α in nuclear fractions; this information has been added (line 318).

13. Line 352: the word "fragmentary" should probably be replaced with "rudimentary".

This has been performed (line 354).

14. Please use 1350 or 1,350 but not 1.350 to describe the number of patients. I have a similar comment for line 483 and 484.

This has been corrected (page 19).

15. The abbreviation EMSA in the legend for figure 7 should be spelled out.

This has been performed.

Once again, we would like to express our sincere gratitude to the reviewers for their fair and helpful comments and hope that we were able to sufficiently reply to the specific issues raised.

Yours sincerely,

Prof. Dr. Ursula Schlötzer-Schrehardt
Friedrich-Alexander-Universität Erlangen-Nürnberg
Department of Ophthalmology
Schwabachanlage 6, 91054 Erlangen, Germany

REVIEWERS' COMMENTS:

Reviewer #1 (Remarks to the Author):

No further comments

Reviewer #2 (Remarks to the Author):

My concerns have been addressed. I appreciate the authors thoughtful responses and transparent revisions/